# Understanding Differential Transformer Unchains Pretrained Self-Attentions

**Chaerin Kong**[1,2]*     **Jiho Jang**[2]*     **Nojun Kwak**[2]

[1] TwelveLabs     [2] Seoul National University

chaerin.k.kong@gmail.com

## Abstract

Differential Transformer has recently gained significant attention for its impressive empirical performance, often attributed to its ability to perform noise canceled attention. However, precisely how differential attention achieves its empirical benefits remains poorly understood. Moreover, Differential Transformer architecture demands large-scale training from scratch, hindering utilization of open pretrained weights. In this work, we conduct an in-depth investigation of Differential Transformer, uncovering three key factors behind its success: (1) enhanced expressivity via negative attention, (2) reduced redundancy among attention heads, and (3) improved learning dynamics. Based on these findings, we propose DEX, a novel method to efficiently integrate the advantages of differential attention into pretrained language models. By reusing the softmax attention scores and adding a lightweight differential operation on the output value matrix, DEX effectively incorporates the key advantages of differential attention while remaining lightweight in both training and inference. Evaluations confirm that DEX substantially improves the pretrained LLMs across diverse benchmarks, achieving significant performance gains with minimal adaptation data ($< 0.01\%$).

## 1 Introduction

Transformer-based architectures have emerged as the cornerstone of modern deep learning across multiple domains [74, 22, 65, 13, 67, 9, 34, 40, 11, 20, 41, 33]. With their attention mechanism, transformers effectively model long-range dependencies, leading to significant advances in large language models [73, 5, 48, 70, 1, 9]. However, a growing body of work [39, 50, 52] highlights that these language models struggle with key information retrieval due to inherent *attention noise*.

To address this issue, Differential (DIFF) Transformer [85] introduces differential attention that computes the difference between two attention scores, thereby boosting attention on key tokens while suppressing common noise. Although its strong empirical performance has established it as a promising alternative to standard transformers, how this simple architecture consistently harnesses the differential operation for effective noise cancellation without explicit guidance remains elusive. Moreover, due to the gap in architecture, employing DIFF attention requires training from scratch, which prohibits utilization of open pretrained weights [73, 5, 70, 48, 59, 27, 1] and incurs huge cost.

In this paper, we aim to fill this gap by providing an in-depth analysis of the mechanisms of DIFF Transformer and presenting a method to efficiently integrate its benefits into existing pretrained transformers. Our key observations are threefold. (1) DIFF attention enhances expressivity through negative attention scores. (2) DIFF attention reduces redundancy among its attention heads. (3) DIFF Transformer exhibits improved learning dynamics.

Building on these insights, we present DEX (**D**ifferential **Ex**tension), an efficient framework that injects the strengths of DIFF Transformer into a pretrained LLM without training from scratch.

---

*Equal contributions.

39th Conference on Neural Information Processing Systems (NeurIPS 2025).

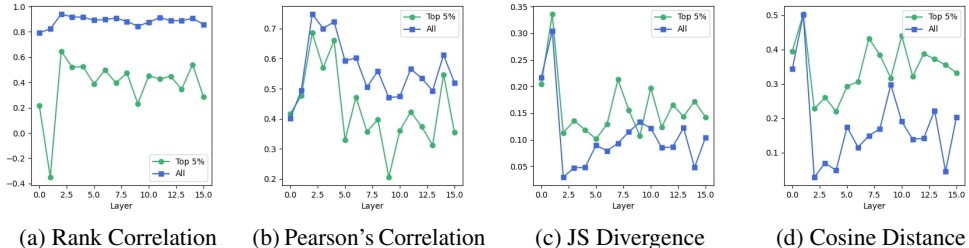

| (a) Rank Correlation | (b) Pearson's Correlation | (c) JS Divergence | (d) Cosine Distance |

Figure 1: Attention score comparison between the two groups in DIFF attention. *Top 5%* refers to top-5% tokens with highest attention score in each sequence. It clearly shows that the overlap between the two attention scores is much greater in non-salient tokens.

Unlike most finetuning methods that fit the model to downstream data, DEX is an architectural adaptation strategy that introduces a key mechanism from a different architecture to a pretrained model, conceptually similar to MHA2MLA [38]. Specifically, DEX operates by reusing the pretrained softmax attention scores (softmax($\mathbf{Q}\mathbf{K}^T$)) and applying its learnable differential mechanism to the output value matrix (softmax($\mathbf{Q}\mathbf{K}^T$)$\mathbf{V}$), making the adaptation lightweight (in both training and inference) yet effective, as demonstrated empirically. To facilitate stable and performant transition, we introduce additional techniques for head selection and $\lambda$-annealing, which controls the critical balance between original knowledge and incoming architectural changes. We validate DEX across multiple model families (Llama-3 [26] and Qwen-2.5 [84]) and scales (0.5B-8B), using diverse benchmarks such as language modeling [25, 77, 79], key information retrieval [39] and in-context learning [6]. DEX consistently achieves significant gains using less than 0.01% the size of the original training data (<1B tokens), without incurring nontrivial test-time overhead.

## 2 How Does Differential Transformer Work?

In this section, we systematically analyze the internal mechanics of DIFF Transformer. Since the original weights are not publicly available at the time of writing, we train a DIFF Transformer on a similar data mix to carry out our analyses. Please refer to Appendix E.2 for full details.

### 2.1 Preliminary: DIFF Transformer

The key innovation of DIFF Transformer is replacing the softmax attentions with DIFF attentions. DIFF attention introduces a mechanism designed to suppress attention noise by computing the difference between attention scores from two separate *groups*. Given an input sequence $X \in \mathbb{R}^{N \times d_{\text{model}}}$, it is first projected into queries, keys, and values as follows:

$$[Q_1; Q_2] = XW_Q, \quad [K_1; K_2] = XW_K, \quad V = XW_V, \tag{1}$$

where $Q_1, Q_2, K_1, K_2 \in \mathbb{R}^{N \times d}$ and $V \in \mathbb{R}^{N \times 2d}$ denote projected matrices, and $W_Q, W_K, W_V \in \mathbb{R}^{d_{\text{model}} \times 2d}$ are learnable parameters. The differential attention is then computed as:

$$\text{DiffAttn}(X) = \left( \text{softmax} \left( \frac{Q_1 K_1^\top}{\sqrt{d}} \right) - \lambda \cdot \text{softmax} \left( \frac{Q_2 K_2^\top}{\sqrt{d}} \right) \right) V, \tag{2}$$

where $\lambda$ is a learnable scalar. This differential mechanism enhances robustness by canceling common-mode attention noise, similar in spirit to differential amplifiers. We note that despite DIFF attention having the same number of parameters, it exhibits significantly higher compute cost and peak memory usage in practice due to enlarged dimensions (see Fig.12). Refer to the original paper [85] for details.

### 2.2 Higher Expressivity via Negative Attentions

The empirical success of DIFF Transformer is often attributed to its *noise-canceling* effect, achieved through subtraction between attention groups. Such noise cancellation is commonly hypothesized to enhance performance by inducing sparsity [85], concentrating attention on relevant context while suppressing irrelevant information. We investigate whether DIFF attention operates primarily through this lens of conventional sparsity [72, 30].

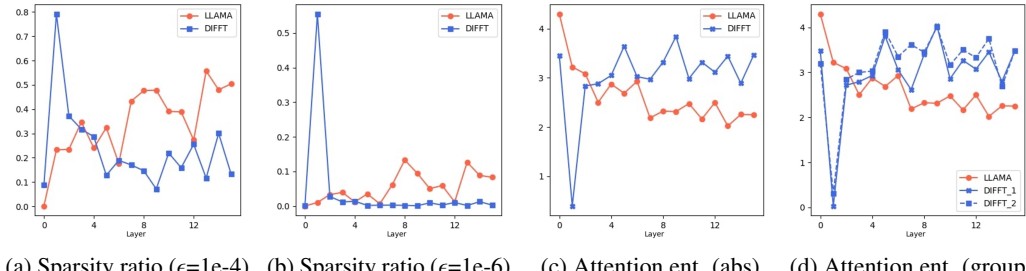

(a) Sparsity ratio ($\epsilon$=1e-4)  (b) Sparsity ratio ($\epsilon$=1e-6)  (c) Attention ent. (abs)  (d) Attention ent. (group)

Figure 2: (a), (b): ratio of attention scores whose absolute value is smaller than $\epsilon$. Except for the bottom layers, DIFF Transformer displays lower sparsity ratio. (c), (d): Attention score entropy. Entropy in (c) measures magnitude concentration, calculated on renormalized absolute values of the final differential attention scores. Group refers to the two separate attentions in DIFF.

Our analysis of DIFF attention's dual attention groups (Fig.1) indeed indicates a form of selective filtering. Metrics such as correlations, Jensen-Shannon divergence [47], and cosine distance between the groups' attention scores (computed pairwise between corresponding heads) reveal high overall similarity (blue) but notably weaker correspondence for the most salient tokens (green). This points to a selective cancellation where shared, less critical attention patterns are offset by the subtraction, while distinct signals for key tokens are largely preserved or emphasized. However, this observed filtering does not directly translate to increased sparsity in its traditional definition (*i.e.,* having many close-to-zero values). In fact, Fig.2(a) and (b) show that DIFF attention often exhibits lower sparsity ratios, while Fig.2(c) and (d) reveal higher entropy values, both indicative of lower sparsity when compared to standard softmax attention.

This suggests that DIFF attention's noise canceling embodies a more nuanced mechanism than simply zeroing out non-salient contexts. As Fig.3 shows, DIFF attention assigns negative scores to a substantial fraction of context tokens, whose relative attention magnitude generally increases in higher layers. Hence, DIFF attention does not uniformly zero out irrelevant contexts, but is capable of flexibly contextualizing them using these signed scores. As [53] shows, employing negative attention to explicitly model *negative relevance* in the query-key (QK) circuit provides greater flexibility to the output-value (OV) matrix, reducing its need for implicit information filtering and thereby fostering more expressive representations. Qualitative examples in Fig.4, such as down-weighting irrelevant subject in Indirect Object Identification task [80] or non-literal interpretation in sarcasm detection, illustrate how DIFF attention can achieve a more refined information flow using negative attention (green boxes). This contrasts with standard softmax attention that assigns high scores even to these highly irrelevant contexts, whose sign-insensitivity often burdens its OV matrix with implicit information filtering [53]. (Additional examples are in Appendix B.5).

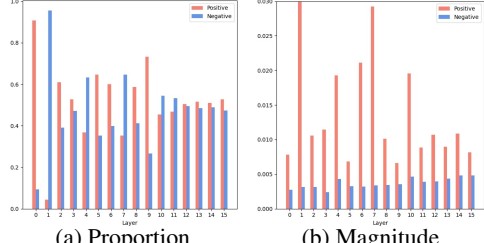

(a) Proportion  (b) Magnitude

Figure 3: The proportion and average magnitude of positive/negative attention scores.

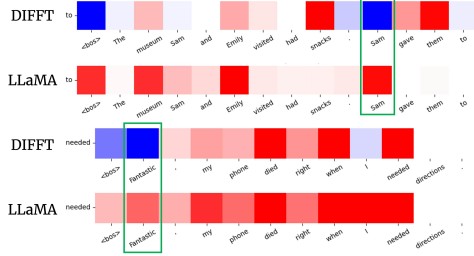

Figure 4: Attention scores on Indirect Object Identification (IOI, top two) and sarcastic expression (bottom two). Blue indicates negative and red represents positive. Green boxes highlight the difference.

## 2.3 Reduced Redundancy among Attention Heads

Multi-head self-attention is powerful but can be redundant [75, 21, 45, 83, 57, 7]. Our analysis reveals that DIFF attention significantly reduces redundancy among attention heads. Fig.5 presents cosine distance between per-head attention scores (higher distance relates to lower redundancy) and Centered Kernel Alignment [61] between value-projected head features (higher alignment translates to higher redundancy). The plots clearly indicate that DIFF attention exhibits reduced redundancy at both the

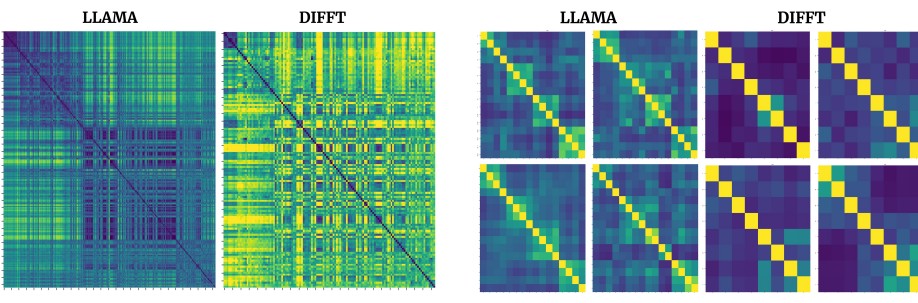



(a) Cosine distance between attention scores.

(b) CKA between attention head features.



Figure 5: (Left) Pairwise cosine distance between per-head attention scores (flattened across layers) Brighter indicates larger distance, hence *less* redundancy. (Right) CKA [61] between per-head features. Brighter means higher alignment, hence *higher* redundancy. See Appendix B.2.

attention score (left) and feature (right) levels. One might attribute this to DIFF having fewer effective heads. However, our experiments demonstrate that merely employing fewer, wider attention heads does not alleviate redundancy (see Appendix B.2). We hypothesize that the differential mechanism grants greater flexibility in controlling attention patterns, reducing inter-head redundancy.

Examining attention head importance provides further insights into head utilization. Fig.6 demonstrates the head importances [60, 15], normalized by the maximum value and sorted. In DIFF Transformer, importance is distributed more uniformly across attention heads (Fig.6 blue), indicating that each head contributes more evenly to the final representation. Combined with the reduced redundancy, this balanced contribution allows DIFF attention to capture a broader spectrum of diverse features compared to conventional multi-head attention.

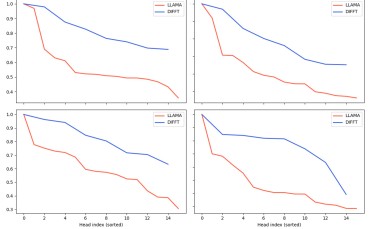

Figure 6: Layerwise head importance distributions, normalized and sorted.

## 2.4 Improved Learning Dynamics

DIFF attention introduces novel components, including the differential operation and a learnable parameter $\lambda$. To understand their impact on learning dynamics, we analyze the Hessian maximum eigenvalue spectra (Fig.7), following the procedure of [64]. As discussed in [64], a high prevalence of negative eigenvalues indicates non-convexity in the loss landscape, which can hinder training, particularly during early phases [63, 19, 36, 35]. We observe significantly fewer negative eigenvalues for DIFF Transformer compared to the standard transformer, suggesting improved optimization dynamics. Notably, this benefit is largely lost when the learnable $\lambda$ is removed (green line in Fig.7).

Training statistics further corroborate this finding. Fig.8 plots the language modeling loss and gradient norms for the standard and DIFF transformer. While DIFF consistently achieves lower loss and more stable grad norms, removing the learnable $\lambda$ notably impairs optimization. We hypothesize that the learnable $\lambda$ plays a key role in stabilizing training dynamics, especially during the early stages.



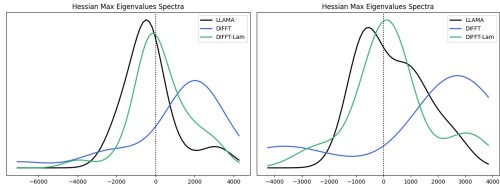

Figure 7: **Hessian max eigenvalue spectra**. While transformer and DIFF without learnable $\lambda$ (DIFFT-Lam) shows a number of negative eigenvalues, DIFF has much less.

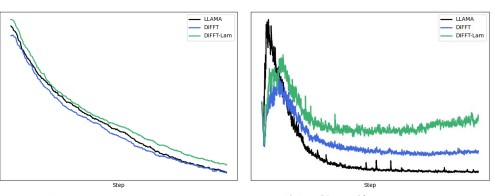

(a) Loss curve. (b) Gradient norm.

Figure 8: **Loss and gradient norm.** DIFF shows the best dynamic while DIFFT-Lam, DIFF with non-learnable $\lambda$, shows instability.



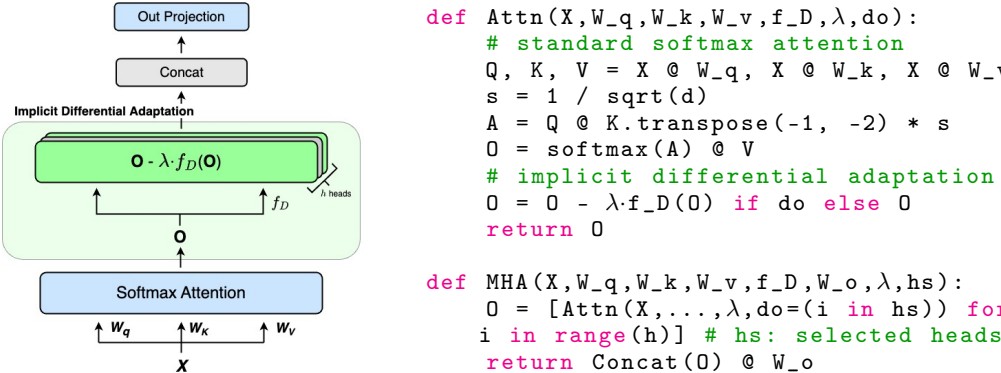

```
def Attn(X,W_q,W_k,W_v,f_D,λ,do):
    # standard softmax attention
    Q, K, V = X @ W_q, X @ W_k, X @ W_v
    s = 1 / sqrt(d)
    A = Q @ K.transpose(-1, -2) * s
    O = softmax(A) @ V
    # implicit differential adaptation
    O = O - λ·f_D(O) if do else O
    return O

def MHA(X,W_q,W_k,W_v,f_D,W_o,λ,hs):
    O = [Attn(X,...,λ,do=(i in hs)) for
    i in range(h)] # hs: selected heads
    return Concat(O) @ W_o
```

Figure 9: **Differential Extension (DEX)**. The output value matrix **O** is transformed by subtracting a $\lambda$-modulated projection from itself. This operation targets a layer-specific subset of attention heads.

## 3 Differential Extension

Based on the insights from Sec.2, we present DEX, a framework that integrates differential mechanism into pretrained self-attentions. In designing DEX, we identify three primary desiderata: (1) effectively integrating the beneficial properties of DIFF Transformer; (2) ensuring a lightweight transition by maximally preserving and leveraging the pretrained knowledge; and (3) minimizing test-time computational or memory overhead. In the following subsections, we describe each component of our framework in detail, explicitly connecting the lessons learned from Sec.2 to satisfy these desiderata.

### 3.1 Implicit Differential Adaptation

Our analysis (Sec.2.2) suggests that DIFF attention's ability to model negative relevance in its QK circuit enhances representational power by facilitating more nuanced information processing in the OV matrix. While this is achieved in DIFF attention by explicitly subtracting two attention scores, naively retrofitting such a dual-group structure onto pretrained models can be problematic. Splitting existing heads into two groups risks significant knowledge loss and instability; duplicating them incurs prohibitive computational and parameter overhead. With DEX, we aim to achieve similar enhancements in information processing, but stably and efficiently.

DEX introduces its learnable differential mechanism directly to the attention *output* instead of the query-key (QK) circuit, an approach we term *implicit* adaptation. This strategy is motivated by the reusability of pretrained attention magnitude signals, supported by our empirical findings that the *absolute* scores of DIFF attention often mirror standard softmax scores (Fig.4, Appendix B.1). By targeting the OV matrix, which is known to control information flow and perform implicit filtering (Sec.2, [53, 23]), DEX empowers the pretrained attention with improved processing of standard attention patterns.

Formally, our implicit differential adaptation is defined as follows:

$$\mathbf{O} = \text{softmax}\left(\frac{\mathbf{Q}\mathbf{K}^\top}{\sqrt{d}}\right)\mathbf{V}, \quad \mathbf{O}' = \mathbf{O} - \lambda f_D(\mathbf{O}), \tag{3}$$

where $f_D$ denotes a learnable projection parameterized by $\mathbf{W}_D$ and $\lambda$ is a learnable scalar. This design offers several notable advantages, including lightweight adaptation through effective knowledge reuse, minimal parameter and test-time compute overhead, and high compatibility with existing transformers. We empirically demonstrate that despite being implicit, DEX effectively delivers the empirical strengths of differential attention.

### 3.2 Selective Adaptation

Attention heads in standard multi-head attention can be highly redundant, and their contribution to the final representation is seldom equal [75, 57]. Further motivated by our findings on effective head utilization in differential attention (Sec.2.3), we propose to *leverage* this inherent heterogeneity

via selective adaptation, applying the implicit adaptation (Eq. 3) only to a subset of heads within each layer, typically targeting those identified as less critical. This selective approach enhances underutilized heads while preserving critical ones, thereby improving overall representational capacity and safeguarding vital pretrained knowledge. We introduce two data-driven head selection strategies:

**Low-Importance Head Selection.** The first method selects heads based on low representational importance, following headwise importance criteria established in [60, 15]. We compute importance scores and apply differential adaptation to the top-$k$ heads with the lowest scores in each layer.

**High-Entropy Head Selection.** The second strategy targets attention heads with high entropy, a state often associated with weaker representational focus, reduced functional specialization, or potential under-utilization [89, 37, 55, 43]. Similarly, we select and adapt the top-$k$ heads demonstrating the highest entropy within each layer.

### 3.3  Balancing Adaptation with Pretrained Knowledge via $\lambda$-Annealing

Our analysis in Sec.2.4 reveals that adaptive modulation of the differential mechanism is critical for stable optimization. In our scenario, maintaining a careful balance between pretrained knowledge and newly introduced architectural modifications is crucial. Zero-initializing the learnable $\lambda$ would be a typical way to safely introduce DEX [32, 88, 28], but that alone does not sufficiently encourage the model to adopt the differential mechanism, as $\lambda$ could remain near zero if the pretrained model is already strong. To facilitate a stable and effective transition, we propose a scheduled annealing of $\lambda$:

$$\lambda(t) = (1 - \alpha) \left[ \frac{t}{T} \lambda_{\text{init}} \right] + \alpha \lambda_{\text{learn}}, \quad \alpha = \min \left( 1, \frac{t}{T} \right) \tag{4}$$

where $t$ is the current training step, $T$ is the annealing duration, $\lambda_{\text{init}}$ is a constant, and $\lambda_{\text{learn}}$ is a learnable parameter initialized around zero. This schedule initiates $\lambda(t)$ with zero for stability, uses annealed $\lambda_{\text{init}}$ to provide a gradual learning signal for the differential mechanism (e.g., $\mathbf{W}_D$) when $0 < t < T$, and transitions control to the learnable $\lambda_{\text{learn}}$ for optimal adaptation as $t \geq T$.

### 3.4  Overall Framework

The complete formulation of DEX for a given head $h$ is expressed as follows:

$$\mathbf{O} = \text{softmax} \left( \frac{\mathbf{Q}\mathbf{K}^\top}{\sqrt{d}} \right) \mathbf{V}, \quad \mathbf{O}' = \mathbf{O} - \lambda(t) \mathbb{I}(h \in \mathcal{H}) f_D(\mathbf{O}), \tag{5}$$

where $\mathcal{H}$ is the set of heads selected for differential adaptation, and $\mathbf{O}'$ is concatenated across all heads and passed into the output projection. During training, we update $\mathbf{W}_\mathbf{K}, \mathbf{W}_\mathbf{V}$, and $\mathbf{W}_\mathbf{O}$ along with $\mathbf{W}_\mathbf{D}$ and $\lambda_{learn}$ within self-attention, keeping all other parameters (*e.g.,* FFN) frozen. This targeted update strategy provides the necessary flexibility to integrate DEX into standard transformers, while keeping the training lightweight, especially under standard GQA [2] setting.

## 4  Experiments

DIFF Transformer has demonstrated strong performance across a wide variety of tasks, including general language modeling, key information retrieval, and in-context learning. We quantitatively validate the effectiveness of DEX in integrating these strengths into pretrained LLMs. We conduct ablation experiments and analyses to further verify our design choices.

### 4.1  Language Modeling Evaluation

**Setup** We apply DEX to Llama-3.1-8B [26], Llama-3.2-3B/1B [56], and Qwen-2.5-1.5B/0.5B [84]. As the original pretraining data for these models is unavailable, we build a custom corpus of web pages, papers, encyclopedias, and code from open datasets [44, 82], similar to OLMo [62]. This corpus contains 887M tokens (Llama-3 tokenizer), less than 0.01% of the models' original pretraining data size. Although DEX is not presented as a fine-tuning method, we compare against baselines trained on the *same* data—including parameter-efficient tuning (PEFT; LoRA [32], PiSSA [54]) and full fine-tuning (FT; Galore [90], APOLLO [93])—to control for the influence of this corpus. Direct

Table 1: Language modeling benchmark scores across model variants and training methods. Green indicates improvement over the baseline, while gray indicates a decrease.

| Model | Arc-C | Arc-E | BoolQ | COPA | Hellaswag | MNLI | OBQA | PIQA | WIC | Winogrande | WSC | AVG | Δ |
|---|---|---|---|---|---|---|---|---|---|---|---|---|---|
| *Llama-3B* | 46.3 | 71.7 | 73.1 | 85.0 | 73.6 | 35.0 | 43.2 | 77.5 | 49.8 | 69.1 | 37.5 | 60.2 | - |
| LoRA (r=8) | 43.4 | 70.2 | 75.3 | 82.0 | 74.2 | 54.5 | 43.0 | 77.1 | 53.8 | 70.1 | 36.5 | 61.8 | +1.6 |
| LoRA (r=32) | 43.7 | 72.0 | 76.2 | 83.0 | 74.7 | 46.7 | 43.2 | 77.7 | 55.2 | 70.0 | 36.5 | 61.7 | +1.5 |
| PiSSA | 45.4 | 73.8 | 74.1 | 82.0 | 74.3 | 46.6 | 42.4 | 78.3 | 56.1 | 69.9 | 38.5 | 61.9 | +1.7 |
| FT | 45.7 | 73.7 | 73.8 | 84.0 | 74.7 | 38.5 | 41.4 | 78.0 | 55.3 | 70.7 | 40.4 | 61.5 | +1.3 |
| GaLore | 46.1 | 74.9 | 76.2 | 87.0 | 74.1 | 33.1 | 42.6 | 77.9 | 53.0 | 70.2 | 38.5 | 61.2 | +1.0 |
| APOLLO | 45.8 | 74.4 | 73.5 | 84.0 | 74.7 | 35.0 | 42.8 | 77.5 | 56.1 | 70.2 | 45.2 | 61.7 | +1.5 |
| Ours | 45.5 | 73.3 | 74.8 | 84.0 | 74.1 | 49.5 | 42.6 | 78.2 | 51.9 | 69.1 | 63.5 | **64.2** | **+4.0** |
| *Llama-1B* | 36.3 | 60.6 | 63.4 | 77.0 | 63.6 | 36.0 | 37.2 | 74.5 | 48.6 | 59.9 | 42.3 | 54.5 | - |
| LoRA (r=8) | 34.6 | 63.3 | 46.4 | 78.0 | 64.1 | 32.9 | 36.6 | 75.1 | 47.9 | 60.9 | 40.4 | 52.7 | -1.8 |
| LoRA (r=32) | 35.9 | 65.4 | 61.5 | 78.0 | 64.4 | 32.6 | 38.2 | 75.1 | 48.7 | 60.3 | 37.5 | 54.3 | -0.2 |
| PiSSA | 36.3 | 65.2 | 59.8 | 79.0 | 64.2 | 33.1 | 37.2 | 75.1 | 49.7 | 60.7 | 38.5 | 54.4 | -0.1 |
| FT | 36.8 | 65.5 | 60.7 | 76.0 | 64.5 | 41.2 | 38.2 | 74.9 | 49.2 | 60.7 | 36.5 | 54.9 | +0.4 |
| GaLore | 36.3 | 65.7 | 60.2 | 77.0 | 64.2 | 34.4 | 37.6 | 75.2 | 50.6 | 60.7 | 36.5 | 54.4 | -0.1 |
| APOLLO | 37.1 | 65.0 | 58.1 | 77.0 | 64.4 | 37.5 | 36.8 | 74.9 | 51.6 | 60.4 | 36.5 | 54.5 | +0.0 |
| Ours | 35.2 | 64.2 | 57.8 | 79.0 | 64.0 | 38.0 | 38.0 | 75.0 | 51.9 | 60.6 | 48.1 | **55.6** | **+1.1** |
| *Qwen-1.5B* | 45.1 | 72.2 | 72.8 | 83.0 | 67.8 | 52.6 | 40.6 | 76.0 | 53.0 | 63.5 | 57.7 | 62.2 | - |
| LoRA (r=8) | 43.3 | 70.3 | 73.5 | 84.0 | 67.5 | 49.3 | 39.2 | 75.1 | 53.3 | 64.3 | 51.0 | 61.0 | -1.2 |
| LoRA (r=32) | 43.4 | 70.2 | 71.0 | 85.0 | 67.5 | 50.7 | 39.2 | 75.5 | 52.0 | 64.7 | 47.1 | 60.6 | -1.6 |
| PiSSA | 44.3 | 70.1 | 72.6 | 84.0 | 66.7 | 47.5 | 40.0 | 74.3 | 54.7 | 63.9 | 52.9 | 61.0 | -1.2 |
| FT | 43.9 | 71.9 | 68.7 | 84.0 | 67.6 | 51.5 | 40.2 | 75.7 | 53.6 | 64.5 | 48.1 | 60.9 | -1.3 |
| GaLore | 44.3 | 72.7 | 72.0 | 84.0 | 67.4 | 47.6 | 39.6 | 75.0 | 53.1 | 64.7 | 51.9 | 61.1 | -1.1 |
| APOLLO | 45.1 | 73.4 | 72.4 | 83.0 | 67.7 | 50.1 | 39.4 | 75.7 | 53.9 | 64.8 | 43.3 | 60.8 | -1.4 |
| Ours | 45.3 | 74.1 | 70.1 | 84.0 | 67.8 | 50.2 | 40.8 | 76.4 | 53.3 | 63.2 | 61.6 | **62.4** | **+0.2** |
| *Qwen-0.5B* | 31.8 | 58.7 | 62.3 | 74.0 | 52.2 | 38.3 | 35.4 | 69.9 | 49.2 | 56.2 | 41.3 | 51.8 | - |
| LoRA (r=8) | 34.3 | 66.1 | 57.2 | 74.0 | 52.3 | 33.9 | 33.6 | 69.4 | 50.0 | 56.2 | 43.3 | 51.8 | +0.0 |
| LoRA (r=32) | 33.4 | 63.9 | 60.6 | 73.0 | 52.1 | 39.1 | 34.4 | 69.7 | 49.2 | 55.6 | 36.5 | 51.6 | -0.2 |
| PiSSA | 34.6 | 66.7 | 59.6 | 73.0 | 51.7 | 33.3 | 33.4 | 69.4 | 50.2 | 56.3 | 36.5 | 51.3 | -0.5 |
| FT | 35.5 | 65.6 | 60.4 | 74.0 | 52.3 | 37.4 | 34.0 | 70.1 | 50.8 | 56.7 | 36.5 | 52.1 | +0.3 |
| GaLore | 35.2 | 65.3 | 58.2 | 74.0 | 52.2 | 34.4 | 33.6 | 70.2 | 49.7 | 56.4 | 36.5 | 51.4 | -0.4 |
| APOLLO | 35.3 | 65.7 | 58.0 | 72.0 | 52.3 | 34.7 | 34.0 | 70.2 | 50.3 | 57.0 | 36.5 | 51.5 | -0.3 |
| Ours | 34.8 | 65.2 | 56.5 | 73.0 | 52.3 | 40.1 | 35.4 | 70.1 | 51.6 | 57.6 | 61.5 | **54.4** | **+2.6** |
| *Llama-8B* | 53.6 | 81.1 | 82.1 | 87.0 | 79.0 | 49.7 | 45.0 | 81.3 | 51.9 | 73.3 | 59.6 | 67.6 | - |
| LoRA (r=8) | 53.1 | 79.5 | 78.3 | 89.0 | 80.4 | 62.1 | 44.8 | 80.5 | 57.8 | 74.9 | 54.8 | 68.7 | +1.1 |
| FT | 52.3 | 80.2 | 80.5 | 91.0 | 80.6 | 60.8 | 45.6 | 81.1 | 58.8 | 73.7 | 57.7 | 69.3 | +1.7 |
| Ours | 52.1 | 79.5 | 79.6 | 91.0 | 80.4 | 58.6 | 46.4 | 80.5 | 58.3 | 75.2 | 64.4 | **69.6** | **+2.0** |

comparison with original DIFF Transformer is limited by unavailable pretrained weights, and we defer evaluations in smaller settings to Appendix along with other details. For head selection we simply set $k$ to be half the total number of heads for each model, and we adopt the $\lambda_{init}$ from [85] (we provide ablations in Appendix B.3). For PiSSA we report $r = 32$ case as this yields good results.

**Results**   We report performances on 11 widely used language modeling benchmarks [16, 78, 76, 86, 58, 8, 68] using [25]. As shown in Table 1, DEX achieves significant improvements across model sizes and families. Given the discrepancy between our training corpus, original pretraining data and the downstream tasks, it is natural to observe degradation after additional training in some cases. Nevertheless, DEX demonstrates robust performance gains on the majority of benchmarks, even when other methods—all trained on the same corpus—exhibit performance drops. In particular, we attribute DEX's strong performance on WSC to its enhanced anaphora resolution granted by the capacity to model negative relevance for incorrect antecedents, which aligns with our intuitions. Notably, although DEX only updates self-attentions, it consistently outperforms both PEFT and full fine-tuning even when full tuning steadily outperforms PEFT (*e.g.,* Llama-1B).

## 4.2   Key Information Retrieval

Needle-in-a-Haystack test [39] is widely adopted to assess LLM's ability to identify critical information embedded in an extensive context. Following the multi-needle retrieval setting of [49, 69, 85], we place the needle at five distinct depths within the context: 0%, 25%, 50%, 75%, and 100%, accompanied by distracting needles. We note the total number of needles placed in the context as $N$, and the number of target needles actually being queried as $R$. Each combination of depth and context length is assessed using 20 samples.

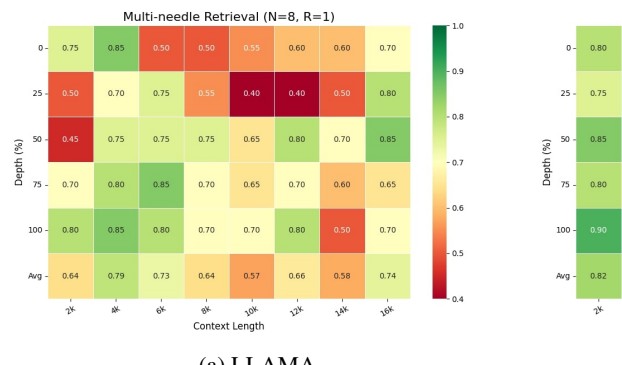
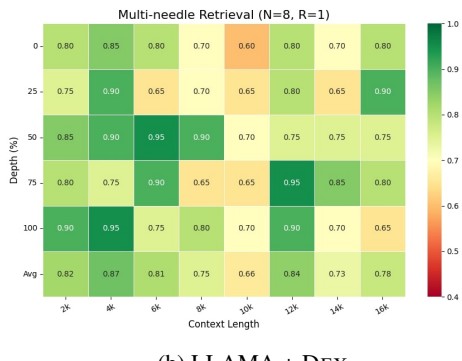

| (a) LLAMA | (b) LLAMA + DEX |
|:---:|:---:|

Figure 10: Multi-needle retrieval results. $N$: total number of needles, $R$: number of queries.

Fig.10 shows the result for $N = 8, R = 1$ case. DEX significantly enhances the retrieval performance of the base Llama-3B model across all context lengths and embedding depths. Notably, DEX improves the average accuracy score by 11.4% (absolute increase from 66.9% to 78.3%), highlighting its effectiveness in improving key information retrieval capabilities.

Increased attention to answer ratio in Table 2 further demonstrates that DEX effectively transfers the core capability of DIFF attention: attention noise canceling. Despite being implicitly applied to the output value matrix, DEX notably alters the *effective attention pattern* to focus on relevant information. This result empirically supports our design choice, placing DEX on the sweetspot between efficiency and efficacy. We provide details in Appendix E.3.

Table 2: Effective attention scores allocated to the answer spans inserted at different depths in key information retrieval.

| | Attention to Answer ↑ | | | | | | |
|---|---|---|---|---|---|---|---|
| Model | 0% | 25% | 50% | 75% | 100% | Avg | |
| Llama | 0.06 | 0.04 | 0.06 | 0.05 | 0.08 | 0.06 | |
| DEX | **0.21** | **0.13** | **0.18** | **0.16** | **0.27** | **0.19** | |

## 4.3 In-Context Learning

DIFF Transformer notably enhances in-context learning performance compared to standard transformer models. To validate whether DEX can achieve similar improvements, we conduct a comprehensive evaluation using three established benchmarks: TREC [31], Banking-77 [12], and Clinic-150 [42]. Following the setup of [6], we adopt a random selection procedure for N-shot examples, as retrieval-based scores quickly saturate with state-of-the-art LLMs.

The results summarized in Table 3 clearly illustrate that DEX consistently delivers performance gains across all evaluated benchmarks compared to both the base Llama model and fine-tuning baselines (LoRA and FT). DEX achieves the highest average accuracy across varying N-shot settings, demonstrating its robustness and efficacy in enhancing the in-context learning capabilities of pretrained models.

Table 3: In-context learning performance.

| | N-shot | | | | | | |
|---|---|---|---|---|---|---|---|
| Dataset | 1 | 10 | 100 | 500 | 1000 | 2000 | Avg |
| *TREC* | | | | | | | |
| Llama | 20.0 | 71.1 | 88.9 | 93.3 | 88.9 | 93.3 | 75.9 |
| LoRA | 20.0 | 68.9 | 93.3 | 93.3 | 91.1 | 93.3 | 76.7 |
| FT | 16.0 | 76.0 | 86.0 | 92.4 | 91.1 | 93.3 | 75.8 |
| DEX | 26.7 | 84.4 | 86.7 | 93.3 | 88.9 | 93.3 | **78.9** |
| *Banking-77* | | | | | | | |
| Llama | 24.4 | 35.6 | 55.6 | 86.7 | 91.1 | 91.1 | 64.1 |
| LoRA | 26.7 | 40.0 | 53.3 | 88.9 | 88.9 | 91.1 | 64.8 |
| FT | 21.6 | 34.4 | 56.0 | 84.4 | 88.8 | 92.4 | 62.9 |
| DEX | 22.2 | 37.8 | 60.0 | 91.1 | 95.6 | 95.6 | **67.0** |
| *Clinic-150* | | | | | | | |
| Llama | 15.6 | 44.4 | 60.0 | 82.2 | 95.6 | 95.6 | 65.6 |
| LoRA | 22.2 | 42.2 | 57.8 | 82.2 | 95.6 | 95.6 | 65.9 |
| FT | 22.2 | 42.2 | 60.0 | 82.2 | 93.3 | 95.6 | 65.9 |
| DEX | 22.2 | 40.0 | 57.8 | 82.2 | 97.8 | 97.8 | **66.3** |

## 4.4 Application to Instruction Tuning

We investigate whether DEX can likewise enhance performance on instruction-following tasks. To fairly assess the effect of DEX, we adopt two complementary settings. First, we apply DEX on a publicly available instruction-tuned checkpoint trained on an open-source instruction corpus OpenHermes-2.5 [71][2] using the same training data (OH-2.5). This *continued* instruction tuning setting eliminates the confounding effect of training data and lets us verify whether DEX improves

---

[2]https://huggingface.co/artificialguybr/Meta-Llama-3.1-8B-openhermes-2.5

Table 4: **Instruction-tuning results on 8 benchmarks.** The top four rows correspond to the first setting, while the bottom two rows correspond to the second.

| Model | MMLU | Arc-C | IFEval | MBPP++ | GSM8K | AGIEval | HumanEval | Math500 | AVG | Δ |
|-------|------|-------|--------|--------|-------|---------|-----------|---------|-----|---|
| *Instruction-tuned* | | | | | | | | | | |
| Base | 62.9 | 78.3 | 46.8 | 68.3 | 71.1 | 32.2 | 44.5 | 13.4 | 52.2 | - |
| + LoRA | 63.1 | 79.5 | 45.7 | 65.3 | 70.3 | 40.6 | 47.0 | 4.0 | 51.9 | -0.3 |
| + FT | 63.0 | 78.6 | 49.2 | 63.2 | 68.8 | 42.3 | 36.6 | 20.0 | 53.7 | +1.5 |
| + DEX | 63.1 | 77.7 | 57.2 | 64.8 | 74.3 | 40.7 | 47.6 | 19.2 | 55.6 | **+3.4** |
| *Pretrained* | | | | | | | | | | |
| + LoRA | 63.7 | 70.5 | 42.0 | 65.3 | 57.4 | 35.4 | 45.7 | 2.0 | 47.8 | -4.4 |
| + DEX | 63.6 | 77.3 | 51.0 | 66.1 | 68.4 | 37.9 | 50.7 | 16.2 | 53.9 | **+1.7** |

the performance of an existing instruct model. Second, we directly apply DEX to a base pretrained model as an instruction-tuning method itself, similarly using OH-2.5 but in single stage. We examine if DEX can effectively induce instruction-following capabilities without prior end-to-end instruction tuning. Note that we include FT (further fine-tuning the open-source checkpoint on the same OpenHermes data for more steps) as an additional baseline to alleviate the concern for underfitting, which clearly distinguishes the contribution of DEX from the benefit of more training steps.

Table 4 reports results on eight representative benchmarks that span language understanding [29], commonsense reasoning [16], instruction following [92], math [17, 29], code generation [4, 14], and general human task [91]. We observe that DEX delivers favorable results on diverse settings, significantly outperforming baselines on benchmarks like GSM8K, HumanEval and IFEval. When directly applied to a base pretrained model, DEX achieves notably higher performance than LoRA, demonstrating comparable performance to more heavily tuned baselines (top 3 rows) without any end-to-end SFT. These results indicate DEX's effectiveness in inducing and reinforcing instruction-following capabilities efficiently.

## 4.5 Ablation and Analysis

We conduct ablation experiments using Llama-3B model. We mainly focus on two critical components: head selection strategies and learnable lambda annealing. We report the average score for the 11 language modeling benchmarks (similar to Table 1). Appendix B presents full results.

From Table 5, it is evident that incorporating entropy-based head selection combined with both learnable and annealed lambda methods yields the best performance, achieving the overall accuracy of 64.2%. Removing either component from lambda leads to noticeable performance drops, indicating the necessity of both. Additionally, both head selection strategies outperform the configuration without head selection, with the entropy-based strategy pushing the boundary further. The fact that choosing low entropy heads (↓) under-

Table 5: Ablation with head selection and lambda control strategies. **imp.** refers to importance-based and **ent.** stands for entropy-based.

| Model | Head Selection | λ-learned | λ-annealed | LM Acc (%) |
|-------|----------------|-----------|------------|------------|
| Llama | - | - | - | 60.2 |
| DEX | all | ✓ | ✓ | 61.9 |
| DEX | imp. | ✓ | ✓ | 63.9 |
| DEX | ent. (↓) | ✓ | ✓ | 62.8 |
| DEX | ent. (↑) | ✓ | ✓ | **64.2** |
| DEX | ent. | ✓ | ✗ | 63.8 |
| DEX | ent. | ✗ | ✓ | 63.4 |
| DEX | ent. | ✗ | ✗ | 62.4 |

performs further supports our design. These findings underline the complementary roles of the head selection and lambda annealing mechanisms in maximizing the effectiveness of DEX.

We also analyze the inner working mechanism of DEX. First, we investigate how DEX modifies the original attention output $\mathbf{O}$ via the subtracted term $\Delta = \lambda f_D(\mathbf{O})$. Fig. 11a plots cosine similarity (indicating modification direction, e.g., positive for suppression) against relative norm (modification magnitude). We observe that while the similarity distribution suggests DEX's capacity to both reinforce and suppress features, heads exhibit distinct patterns, with some focusing on amplification (higher norm for negative cosine, left) and others on attenuation (higher norm for positive cosine, right). Second, CKA on head output features reveals that DEX notably reduces inter-head redundancy (Fig.11b), implying more diverse head specialization. Lastly, we monitor $\lambda$ during training in Fig.11c. Simply zero-initializing the learnable $\lambda$ (red) completely fails to introduce DEX, while removing annealing (green) results in instability at the initial phase. Our approach (blue) smoothly introduces DEX with minimal damage to the pretrained knowledge. Refer to Appendix for full results.

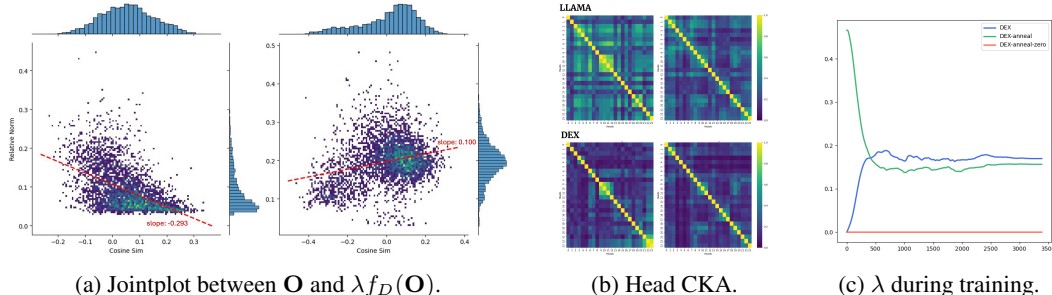

(a) Jointplot between $\mathbf{O}$ and $\lambda f_D(\mathbf{O})$.      (b) Head CKA.      (c) $\lambda$ during training.

Figure 11: **Analysis on DEX**. (a) Cosine similarity $(\mathrm{cosine}(\mathbf{O}, \Delta))$ vs. Relative Norm $(||\Delta||/||\mathbf{O}||$, where $\Delta = \lambda f_D(\mathbf{O}))$ (b) CKA of attention head output features (brighter means higher redundancy). (c) Training dynamics of learnable $\lambda$ under different initialization/annealing schemes.

We verify the test-time efficiency of DEX by comparing the throughput (tokens per second) and latency with base Llama and DIFF Transformer (3B). Fig.12 clearly shows that both in terms of throughput and latency, DEX demonstrates superior inference time efficiency. While DIFF Transformer exhibits increasing inefficiency with longer context due to its compute-heavy attention operation, DEX remains competitive to the original Llama baseline thanks to its lightweight design.

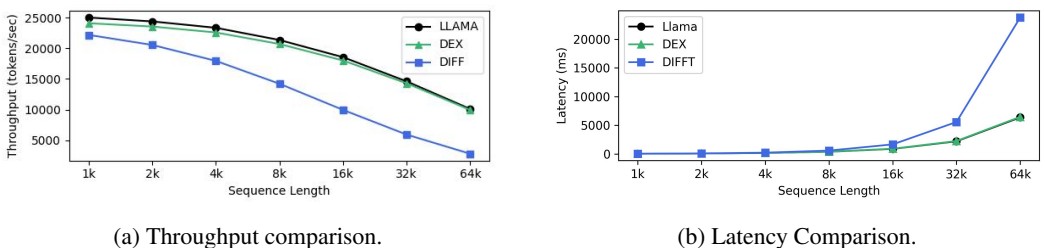

(a) Throughput comparison.      (b) Latency Comparison.

Figure 12: **Inference time efficiency analysis.** We benchmark (a) throughput and (b) latency of three attention variants. While DIFF attention costs significantly more compute at test-time compared to the original Llama attention, DEX incurs negligible overhead thanks to its lightweight design. All benchmarks are measured on a single Nvidia A100 GPU.

Finally, we evaluate the effect of training data size on the application of DEX (Fig. 13). Notable gains appear with just 400M tokens, highlighting the lightweight nature of DEX. Since our training data lacks direct correlation with the downstream benchmarks, modest amount of data (<1B) is sufficient to elicit the full potential of DEX, and simply adding more general data yields diminishing returns in downstream tasks.

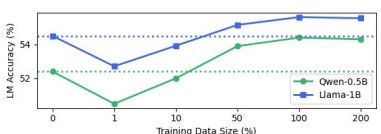

Figure 13: DEX with varying data size.

## 5 Conclusion

In this work, we study the internal mechanism of DIFF Transformer to identify three key factors behind its empirical success: enhanced expressivity via negative attention, reduced redundancy among attention heads and improved optimization dynamics. Based on these insights, we propose DEX, an architectural adaptation method that efficiently integrates the empirical strengths of DIFF Transformer into pretrained LLMs with standard transformer architecture. Diverse evaluation results confirm the effectiveness and versatility of DEX.

**Limitations** For DIFF Transformer analysis, we followed the original setup as much as possible, but different behaviors can emerge under different model scale, data composition, etc. Similarly, though DEX works well across model sizes, it has not been tested beyond 8B parameter scale. We leave it for future works.

**Acknowledgement** N. Kwak was supported by NRF (2021R1A2C3006659) and IITP grants (RS-2021-II211343, RS-2022-II220320, RS-2025-25442338), all funded by the Korean Government.

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

# A Related Works

**Differential Transformer** [85] introduces an architecture designed to mitigate attention noise [39, 50, 52], a known challenge in transformer models. Its core mechanism involves computing attention scores using two groups and then subtracting the resulting attention maps, aiming to cancel out common-mode noise components. Building on this, DINT Transformer [10] aims to improve numerical stability and training dynamics by incorporating an integral term alongside the differential one. However, these pioneering works do not provide detailed mechanistic analyses explaining *why* differential attention is effective. Furthermore, both architectures inherently require computing two separate attention pathways, resulting in substantial computational overhead compared to standard attention. This increased cost hinders practical deployment, particularly for large-scale models. Building on top of our analysis on DIFF Transformer's success, we propose DEX that implicitly embeds the benefits of differential attention into pretrained language models without nontrivial computational overhead.

**Negative Attention Scores.** Several approaches explicitly introduce negative attention scores. Centered Attention [3], for instance, adds offsets to the softmax calculation, forcing attention weights per query to sum to zero (rather than one) to mitigate over-smoothing. Other methods achieve negative weighting through direct manipulations of the softmax operation or via linear attention approximations [53, 55], often demonstrating enhanced representational expressivity. However, methods that explicitly alter the core attention computation can introduce training stability challenges and often lack compatibility with highly optimized implementations like FlashAttention [18]. In contrast, DEX aims to capture the benefits of signed, differential attention implicitly. By applying its learnable transformation *after* the standard softmax attention calculation (i.e., to the output values), DEX avoids modifying the core QK-softmax pathway, thereby maintaining compatibility and potentially simplifying integration and training.

**Attention Redundancy and Entropy.** Numerous works [15, 60, 57, 75, 21, 45, 83, 7] have shown that there is significant redundancy among attention heads in multi-head attention, and propose head pruning methods to enhance efficiency. Instead of getting rid of unimportant heads, our approach applies implicit differential adaptation to redundant heads, effectively revitalizing them and modeling richer attention representations. Meanwhile, attention entropy-based analyses have provided insights into the transformer attention mechanisms. [89, 55] argues that excessively high attention entropy negatively impacts performance, while [37, 87] associates entropy with training stability. In this work, we leverage attention entropy in two ways: (1) understanding the attention score distribution (and potential sparsity), and (2) identifying less critical attention heads.

# B Ablation and Analysis

In this section, we present additional empirical results to support our design choice and analysis.

## B.1 Attention Magnitude Correlation

In Fig.14, we present the correlation between attention scores from Llama and DIFF Transformer, computed layer by layer. Specifically, we compare Llama's softmax scores against both the original signed scores from DIFF attention (green) and their absolute values (blue). Note that while standard Llama attention scores are non-negative (due to softmax), DIFF attention scores can be negative. Interestingly, both rank and Pearson correlations are significantly higher when using absolute values (blue) compared to signed values (green). This suggests strong correspondence in the *magnitude* of attention (indicating relative impor-

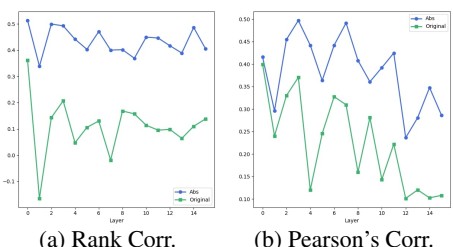

(a) Rank Corr.      (b) Pearson's Corr.

Figure 14: Correlation between Llama attention and DIFF attention.

tance), even when the signed scores differ. This observation motivates our DEX design: since the relative importance signals (magnitudes) from the standard QK/softmax pathway are largely preserved, we reuse them and focus our adaptation efforts on enhancing the subsequent OV circuit to incorporate differential mechanism.

## B.2 Attention Head Redundancy

We address the potential concern that lower inter-head redundancy in DIFF Transformer stems from its common configuration using fewer, wider attention heads (typically halving head count while doubling head dimension [3]).

We plot the average pairwise cosine distance between head attention scores per layer (Fig.15). The figure shows DIFF attention exhibiting significantly higher average cosine distance, indicating lower redundancy (greater pattern dissimilarity) among its heads. Notably, merely using fewer, wider heads does not replicate this effect, as demonstrated by our LLAMA-half baseline (green), configured with halved head count and doubled head dimension. We hypothesize that the differential mechanism grants greater flexibility in controlling attention patterns, thus reducing inter-head redundancy.

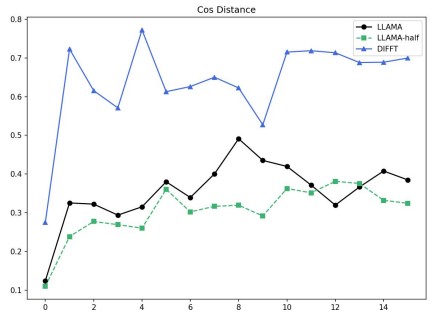

Figure 15: Mean pairwise cosine distance between attention scores from different heads.

Heatmaps visualizing the pairwise cosine distances between attention maps from different heads (Fig.16) further corroborate our findings. They show lower inter-head distances (indicating higher similarity and redundancy) in the standard transformer (LLAMA), whereas DIFF Transformer maintains higher distances, demonstrating more diverse attention patterns.

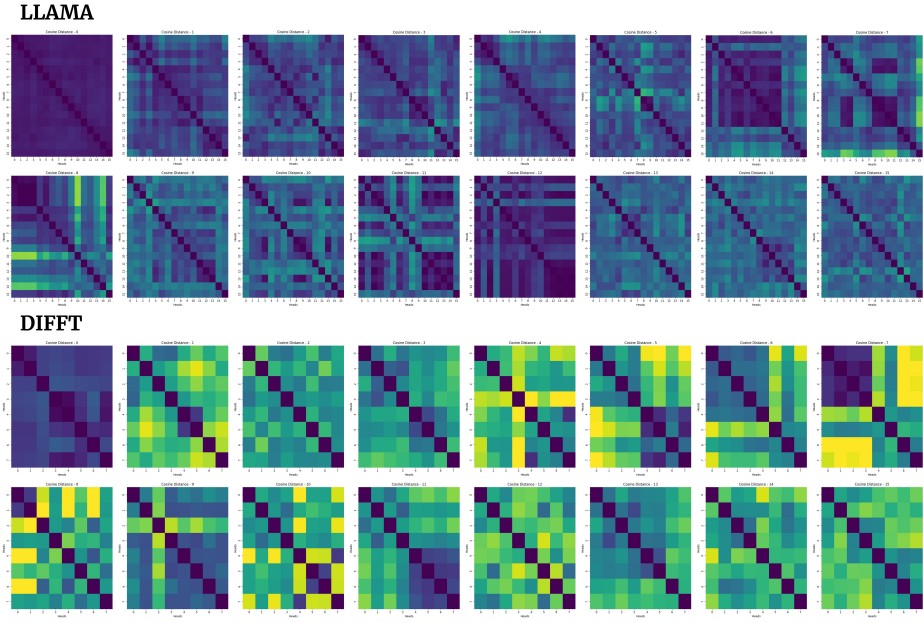

Figure 16: Pairwise cosine distance between attention maps from different attention heads in each layer. Brighter color indicates larger distance, hence lower redundancy.

Centered Kernel Alignment (CKA) analysis comparing attention heads before and after applying DEX (Fig.17) further confirms that DEX reduces inter-head redundancy. The results clearly show lower overall alignment between heads after adaptation in the pretrained models, indicating increased functional diversity.

---

[3] https://github.com/microsoft/unilm/issues/1663

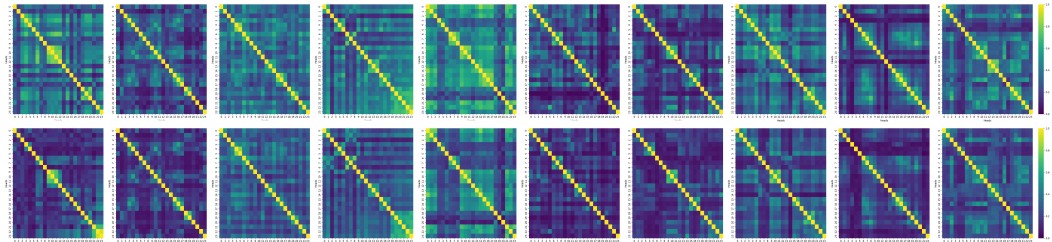

Figure 17: Centered Kernel Alignment for attention heads. Brighter colors indicate higher alignment/similarity. (Top) Llama, (Bottom) DEX.

## B.3 Abalation on $\lambda_{init}$

Table 6 shows DEX performance on the language modeling benchmarks (average over 11 tasks from Table 1, using Qwen-0.5B) when varying the $\lambda_{init}$ strategy. The results indicate relative robustness to different fixed scalar initializations (0.3-0.8). However, adopting the initialization scheme from the original DIFF Transformer setting yields slightly the best

Table 6: Ablation on $\lambda_{init}$. DIFF refers to depth-aware initialization following [85].

| $\lambda_{init}$ | 0.8 | 0.5 | 0.3 | DIFF |
|---|---|---|---|---|
| LM Acc (%) | 54.3 | 54.0 | 54.2 | **54.4** |

performance. We hypothesize the layer-aware initialization is beneficial for training.

## B.4 Ablation on Head Selection $k$

In Table 7, we present the average performance of DEX with different number of target attention heads ($k$) on 11 language modeling benchmarks. Selecting too few heads (*e.g.*, $k = 8$) provides insufficient capacity for the differential adaptation, while modifying too many heads risks disrupting critical pretrained

Table 7: Ablation on head selection $k$.

| $k$ | 8 | 16 | 24 | 32 |
|---|---|---|---|---|
| LM Acc (%) | 53.7 | **55.6** | 54.4 | 54.1 |

knowledge, leading to performance degradation. We empirically find modifying about 50% of the attention heads tends to be optimal in general (note we use Llama-1B with 32 heads for ease of demonstration).

## B.5 Qualitative Results

We display additional examples from qualitative analysis in Sec.2.2.

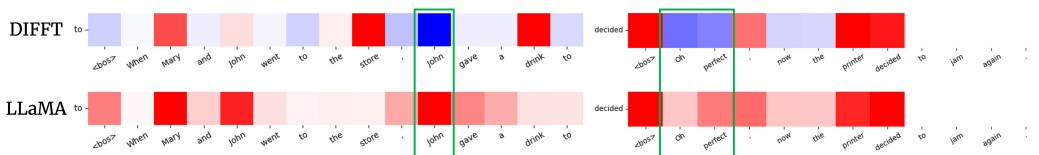

Figure 18: Qualitative examples for DIFF Transformer and Llama.

Fig.18 aligns with the observations from Fig.4, illustrating how DIFF attention leverages negative attention scores. The left example shows an Indirect Object Identification task where DIFF Transformer assigns a negative attention score to mark the subject (*i.e.,* John) as irrelevant. The right example shows sarcasm detection, where DIFF attention identifies the non-literal expression and explicitly allocates negative attention scores accordingly.

## C Comparison to DIFF Transformer

A direct comparison with the original DIFF Transformer model is not possible due to unavailable weights. Therefore, to establish a point of reference, we compare DEX with a DIFF Transformer model

that we trained ourselves at a smaller scale, following the procedures detailed in Appendix E.2. To set up this comparison, we first train two models from scratch on the exact same training data: (1) a standard transformer baseline using the Llama architecture (Llama), and (2) DIFF Transformer model (DIFF). Subsequently, we apply DEX to the Llama baseline using a small subset of the pretraining data (<1B tokens) to create the third model, simply noted DEX. We additionally train a separate Llama model from scratch with DEX attached from the beginning, to understand DEX's architectural capacity beyond its original purpose of adaptation, which we refer to as DEX-S. We report the performance of these four models (Llama, DIFF, DEX, DEX-S) on the 11 language modeling benchmarks in Table 8.

Table 8: Scores on 11 benchmarks. Green indicates increases and gray indicates decreases. All values are rounded to one decimal place.

| Model | Arc-C | Arc-E | BoolQ | COPA | Hellaswag | MNLI | OBQA | PIQA | WIC | Winogrande | WSC | AVG | Δ |
|-------|-------|-------|-------|------|-----------|------|------|------|-----|-----------|-----|-----|---|
| Llama | 21.8 | 37.0 | 60.5 | 63.0 | 29.0 | 35.1 | 25.2 | 58.4 | 50.6 | 49.6 | 36.5 | 42.4 | - |
| DIFF | 24.2 | 37.2 | 54.0 | 68.0 | 29.0 | 35.5 | 26.4 | 58.9 | 50.0 | 52.2 | 36.5 | 42.9 | +0.5 |
| DEX | 22.2 | 37.1 | 60.5 | 64.0 | 29.0 | 35.1 | 25.8 | 58.3 | 50.6 | 51.2 | 36.5 | 42.8 | +0.4 |
| DEX-S | 22.5 | 37.1 | 61.5 | 63.0 | 28.7 | 35.2 | 27.4 | 58.1 | 50.0 | 51.0 | 36.5 | 42.8 | +0.4 |

From the table, we first observe that DIFF Transformer generally outperforms standard transformer on the majority of benchmarks, which supports the strength of DIFF Transformer as a general purpose language model. Furthermore, the results clearly demonstrate that DEX, despite being lightweight both during training and inference, effectively enhances the pretrained Llama model, closing the gap between standard transformer and DIFF Transformer. DEX-S, a variant of DEX applied from scratch, also delivers competitive performances beyond standard Llama model.

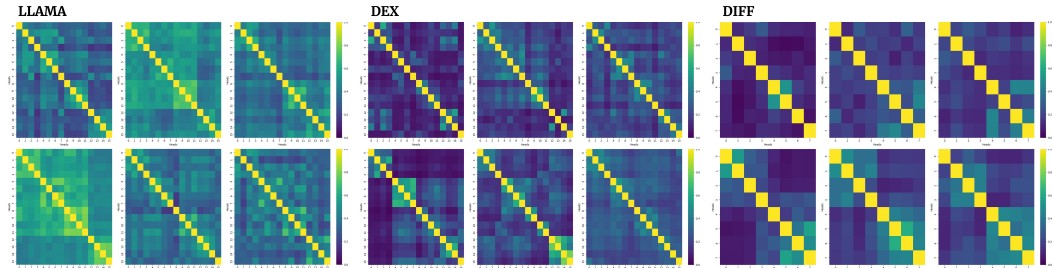

Figure 19: Head CKA comparison between Llama vs DEX vs DIFF.

Head CKA results further support the effectiveness of DEX (Fig.19). Compared to standard transformer (left), DEX significantly reduces the inter-head redundancy (indicated by lower alignment), yielding similar results to DIFF Transformer.

# D  Efficiency Analysis

To evaluate inference efficiency, we benchmark throughput (tokens per second) for 3B-parameter versions of Llama, DIFF Transformer, and DEX, presenting the results in Fig.12a. Context lengths were varied from 1k to 64k tokens to cover a comprehensive range of use cases. All tests were conducted on a single NVIDIA A100-80GB GPU, utilizing PyTorch's standard scaled dot-product attention implementation[4]. The reported throughputs are averaged over 30 batches, following an initial 5 warm-up batches.

# E  Implementation Details

In this section, we provide comprehensive details for our experiments, some of which were abbreviated in the main manuscript for brevity.

---

[4]`https://docs.pytorch.org/docs/stable/generated/torch.nn.functional.scaled_dot_product_attention.html`

### E.1 Language Modeling Evaluation

**Dataset** We constructed our custom training corpus using a subset of the Dolmino dataset[5]. Specifically, we mixed web pages, academic papers, encyclopedia entries, and code texts in approximate ratios of 74.3%, 6.5%, 7.9%, and 11.3% respectively. This resulted in a corpus totaling 887M tokens (measured using the Llama-3 tokenizer). Our data preparation generally followed the recipe of OLMo2 [62], with the main exception being a greater upsampling of the code text component.

**Training** All models, including baselines and DEX variants, were trained on our custom corpus for 1 epoch. A context length of $32k$ tokens was used for all Llama and Qwen models during this training phase. We employed a cosine learning rate schedule, using a peak learning rate of $1 \times 10^{-4}$ for partial fine-tuning methods (including DEX) and $1 \times 10^{-5}$ for full fine-tuning baselines, as these settings generally yielded the best outcomes in preliminary experiments. A learning rate warm-up ratio of 0.03 was used. All experiments were conducted using 8 NVIDIA A100-80GB GPUs, with the run time ranging from 2.5-16 hours depending on the model size.

### E.2 Training DIFF Transformer

We train our own DIFF Transformer model for analysis. This subsection details its training procedure.

**Dataset** We followed the recipe of DIFF Transformer and StableLM-3B[6], using various open-source datasets [66, 24, 46, 81] to create a corpus of approximately 30 billion tokens (Llama-3 tokenizer). This corpus encompasses a diverse range of domains, including academic papers, source code, encyclopedic articles, and literature.

**Model** We trained a 0.4-billion parameter version of DIFF Transformer. Key architectural parameters are provided in Table 9.

Table 9: Configuration for 0.4B DIFF Transformer.

| params | values |
|---|---|
| # Layers | 16 |
| # Heads | 16 |
| # KV Heads | 4 |
| Hidden size | 1024 |
| FFN size | 4096 |

**Training** For training, we employed the AdamW optimizer [51] with a cosine learning rate schedule. The peak learning rate was set to $1 \times 10^{-4}$, the global batch size to 256, and the learning rate warm-up to 0.1. The $\lambda$ parameters within the differential attention were initialized according to the exact schedule specified in the original DIFF Transformer paper [85].

### E.3 Approximating Effective Attention Scores for DEX Interpretability

Because DEX directly alters the attention block's output value matrix $\mathbf{O}$ rather than the initial softmax scores, standard attention visualization can be misleading. To provide insight into its effective learned behavior, we propose methods to approximate the *effective attention scores* that would yield DEX's modified output using the original value matrix.

**Least-Squares Approximation** This method uses the Moore-Penrose pseudoinverse to derive effective attention scores $\mathbf{X}$ that best reconstruct DEX's output transformation. Specifically, let $\mathbf{A} = \text{softmax}(\mathbf{Q}\mathbf{K}^T/\sqrt{d_k})$ be the original softmax attention scores from a given head, $\mathbf{V} \in \mathbb{R}^{N \times d_v}$ be the corresponding value matrix (where $N$ is sequence length, $d_k$ is key dimension, $d_v$ is value dimension), and $\mathbf{W}_D \in \mathbb{R}^{d_v \times d_v}$ be the learnable weight matrix for $f_D$ in DEX (assuming $\lambda(t)$ is

---

[5] https://huggingface.co/datasets/allenai/dolmino-mix-1124
[6] https://github.com/Stability-AI/StableLM

absorbed into $\mathbf{W}_D$ or considered $\approx 1$ for this analysis). The original head output is $\mathbf{O} = \mathbf{AV}$, and the DEX-modified output is $\mathbf{O}' = \mathbf{O}(I - \mathbf{W}_D)$. We seek an effective attention score matrix $\mathbf{X}$ such that $\mathbf{XV} \approx \mathbf{O}'$.

The least-squares solution for $\mathbf{X}$ is:

$$\mathbf{X} = \mathbf{O}'\mathbf{V}^+ = \mathbf{AV}(I - \mathbf{W}_D)\mathbf{V}^+ \tag{6}$$

where $\mathbf{V}^+$ denotes the Moore-Penrose pseudoinverse of $\mathbf{V}$, computed numerically in practice.

This $\mathbf{X}$ represents the attention pattern that, if applied to the original values $\mathbf{V}$, would best reconstruct DEX's modified output for that head. Since this involves an approximation and the use of a pseudoinverse (which can be sensitive if $\mathbf{V}$ is ill-conditioned or has a significant null space), numerical considerations are important. We therefore complement and cross-check these results using a second technique.

**Optimization-based Approximation**    As with the pseudoinverse method, we aim to find an effective attention score matrix $\mathbf{X}$ such that $\mathbf{XV}$ approximates the DEX output $\mathbf{O}' = \mathbf{AV}(I - \mathbf{W}_D)$. Rather than a closed-form pseudoinverse solution, this approach directly optimizes for $\mathbf{X}$ for each input sample for which $\mathbf{O}'$ and $\mathbf{V}$ are computed. For each sample, $\mathbf{X}$ is typically initialized (e.g., as the original attention scores $\mathbf{A}$) and then updated for 100 iterations using gradient descent (learning rate $1 \times 10^{-3}$) to minimize a reconstruction loss with the form $||\mathbf{XV} - \mathbf{O}'||_2^2$.

The primary interpretable attention scores reported in our main analyses (e.g., Table 2) were derived using the pseudoinverse method. This optimization-based approach served as a cross-validation, and we confirmed strong agreement between the effective attention scores obtained from both techniques. While both methods yield approximations subject to numerical precision, they offer valuable tools for understanding the internal mechanisms and effective attention patterns of DEX.

# F   Broader Impact

**Potential Positive Societal Impacts:** By improving core LLM capabilities such as information retrieval, in-context learning, and overall representational quality, DEX could contribute to more effective and reliable AI systems. This includes advancements in AI-assisted education, more capable research tools, improved accessibility to information, and more helpful AI assistants. Furthermore, DEX's design emphasizes lightweight adaptation, which could make powerful LLM enhancements more resource-efficient and accessible, potentially reducing the computational burden associated with adapting large models.

**Potential Negative Societal Impacts:** As DEX is designed to improve the capabilities of LLMs, it shares the potential negative societal impacts inherent in more powerful language model technology. Enhancements in LLM performance and efficiency could inadvertently facilitate the creation of more sophisticated or scalable misuse scenarios, such as generating convincing disinformation, spam, or impersonations. If an LLM enhanced by DEX produces incorrect or biased information, its improved fluency might make such outputs seem more authoritative, potentially exacerbating harm. While DEX is a foundational architectural improvement rather than a specific end-user application, the dual-use nature of advancements in LLM capabilities warrants careful consideration.

We believe that continued research into robust AI safety measures, ethical development guidelines, bias detection and mitigation, and responsible deployment practices for all LLMs is crucial as their capabilities, including those enhanced by methods like DEX, advance.

