# OpenReview forum: "Understanding Differential Transformer Unchains Pretrained Self-Attentions"
_NeurIPS.cc/2025/Conference — NeurIPS 2025 poster_

### Official Review · Reviewer_HPZ6 · 2025-06-29

**Clarity:** 4
**Significance:** 3
**Originality:** 3
**Rating:** 4
**Confidence:** 4

**Summary:**

This paper presents two main contributions. Firstly, it provides an investigation into Differential Transformers, including their internal workings, the phenomenon of Negative Attention, reductions in attention head redundancy, and their training dynamics. Secondly, the authors introduce DEX, a transformer architecture that, while conceptually related to Differential Transformers, is still different. DEX offers a one advantage over Differential Transformers: its applicability to pretrained models. Through extensive experimentation on various pretrained transformer models, the authors demonstrate DEX's superior performance across multiple tasks, outperforming LoRA, full fine-tuning, and other common fine-tuning strategies. The paper is well-supported by a thorough ablation study and offers valuable initial insights into the characteristics of models fine-tuned with DEX.

**Questions:**

Q1: Regarding Figure 8a, the trend seems to indicate that training for more steps benefits LLaMA more than the Differential Transformer, potentially leading to a better final loss for LLaMA. Could the authors train for a longer duration and report these extended results to explore whether a regular Transformer architecture might ultimately be superior?

**Ethical Concerns:**

["NO or VERY MINOR ethics concerns only"]

**Final Justification:**

I've reviewed the rebuttal and the discussions with the authors, and my score remains unchanged.

**Limitations:**

Yes

**Quality:**

3

**Strengths And Weaknesses:**

Strengths:

W1: The paper provides detailed investigation of Differential Transformers, illustrating the components learned by this architectural design. This contributes to understanding the inner workings of this architecture.

W2: The proposed DEX architecture is novel and appears to outperform existing fine-tuning approaches.

W3: The ablation study is both important and well-executed.

W4: The paper is clear and well-structured.

Weaknesses

S1: The main weakness here is that the DEX architecture is actually different from Differential Transformers, and it's not truly clear that DEX performs as Differential Transformers do. I think you should first compare Differential Transformer and DEX when trained from scratch, as was done for Section 2 when comparing to LLaMA. Then, assuming DEX and Differential Transformers behave similarly, you can apply DEX to pretrained models. Training DEX from scratch, mirroring how LLaMA and Differential Transformers were handled for Section 2, would be very beneficial.

S2: A more minor weakness is that throughout the paper, all the legends in the figures are very small and very hard to read. The authors should fix this for improved clarity.

---

> ### Author Rebuttal · Authors · 2025-07-30
>
> We deeply appreciate your thoughtful review and constructive feedback. We are pleased to hear that you found our work insightful, novel, well-executed and well-structured. We hereby address your concerns one by one.
>
> **Concern 1: Analyzing DEX-from-scratch as done in Sec.2 to confirm its internal resemblance to Differential Transformer would be beneficial.**
>
> We believe this is a very insightful feedback. To provide some context for our original thinking, we did not believe side-by-side comparison between DEX and Differential Transformer (like in Sec.2) was *necessary* because our primary goal for DEX was not to replicate all the internal mechanisms of DIFF. Rather, it was to efficiently inject the downstream strengths of DIFF (as evidenced by natural language tasks, information retrieval, etc) to pretrained transformer models for practical purpose using our learned insight as design inspiration, though we do believe that there is strong correlation between the "internal mechanism" and the "downstream performance".
>
> That being said, we agree that including the analysis would indeed add clarity and completeness to our claims about DEX. Below we present 3 key analyses introduced in Sec.2, namely (1) attention score correlation (**Fig.1**), (2) negative attention ratio (**Fig.3**), and (3) attention head CKA (**Fig.5**). We ask for your understanding on presenting these results in raw numbers, as communication through image is no longer possible.
>
> | Layer | Correlation (top-5%) | Correlation (all) | Rank Correlation (top-5%) | Rank Correlation (all) |
> | :--- | :--- | :--- | :--- | :--- |
> | 1 | 0.12 | 0.46 | 0.07 | 0.34 |
> | 2 | -0.09 | 0.14 | 0.14 | 0.47 |
> | 3 | -0.07 | 0.22 | 0.12 | 0.36 |
> | 4 | 0.09 | 0.36 | -0.07 | 0.21 |
> | 5 | 0.21 | 0.36 | 0.04 | 0.14 |
> | 6 | 0.24 | 0.36 | 0.09 | 0.16 |
> | 7 | 0.52 | 0.50 | 0.30 | 0.14 |
> | 8 | 0.58 | 0.52 | 0.39 | 0.18 |
> | 9 | 0.17 | 0.28 | -0.03 | 0.14 |
> | 10 | 0.18 | 0.17 | -0.08 | 0.05 |
> | 11 | 0.55 | 0.41 | 0.27 | 0.08 |
> | 12 | 0.10 | 0.37 | -0.26 | -0.05 |
> | 13 | 0.48 | 0.18 | 0.15 | -0.02 |
> | 14 | 0.21 | 0.21 | -0.16 | -0.05 |
> | 15 | 0.18 | 0.19 | -0.21 | -0.05 |
> | 16 | 0.26 | 0.01 | -0.09 | -0.02 |
> | **Average** | **0.23** | **0.30** | **0.04** | **0.13** |
>
> The above shows the correlation (Pearson's) and rank correlation between the two *group*'s attention scores, measured on our 0.4B DEX model trained from scratch. We applied DEX to all attention heads because we were training from scratch anyways, and computed *effective attention scores* for this analysis as detailed in **Appendix F.3**. Note that this experimental setup applies to all three tables presented here.
>
> It is clearly visible from the table that just like in **Fig.1**, DEX-from-scratch shows lower correspondence (overlap) on salient tokens with the highest attention scores, potentially indicating a form of selective filtering of common noise.
>
> Next, we present layer-wise positive/negative ratio for the attention scores of DEX.
>
> | Layer | Positive | Negative |
> | :--- | :--- | :--- |
> | 0 | 0.69 | 0.31 |
> | 1 | 0.61 | 0.39 |
> | 2 | 0.59 | 0.41 |
> | 3 | 0.62 | 0.38 |
> | 4 | 0.56 | 0.44 |
> | 5 | 0.59 | 0.41 |
> | 6 | 0.56 | 0.44 |
> | 7 | 0.59 | 0.41 |
> | 8 | 0.53 | 0.47 |
> | 9 | 0.52 | 0.48 |
> | 10 | 0.52 | 0.48 |
> | 11 | 0.50 | 0.50 |
> | 12 | 0.50 | 0.50 |
> | 13 | 0.51 | 0.49 |
> | 14 | 0.51 | 0.49 |
> | 15 | 0.51 | 0.49 |
>
> This demonstrates that like Differential Transformer (in **Fig.3**), DEX induces a significant proportion of *negative attention scores* once learned. Our qualitative inspection indicates the formation of a "more expressive attention mechanism" thanks to this increased flexibility, as was the case with Differential Transformer (**Sec.2.2**).
>
> Lastly, we show the head-CKA result that quantifies the redundancy between attention heads.
>
> | layer | Llama | DIFFT | DEX |
> | :--- | :--- | :--- | :--- |
> | 0 | 0.46 | 0.51 | 0.39 |
> | 1 | 0.41 | 0.23 | 0.38 |
> | 2 | 0.43 | 0.31 | 0.39 |
> | 3 | 0.45 | 0.38 | 0.37 |
> | 4 | 0.43 | 0.30 | 0.37 |
> | 5 | 0.38 | 0.47 | 0.33 |
> | 6 | 0.45 | 0.43 | 0.32 |
> | 7 | 0.40 | 0.31 | 0.34 |
> | 8 | 0.45 | 0.29 | 0.31 |
> | 9 | 0.56 | 0.33 | 0.35 |
> | 10 | 0.37 | 0.36 | 0.37 |
> | 11 | 0.32 | 0.35 | 0.38 |
> | 12 | 0.44 | 0.39 | 0.33 |
> | 13 | 0.57 | 0.43 | 0.36 |
> | 14 | 0.46 | 0.44 | 0.43 |
> | 15 | 0.62 | 0.52 | 0.47 |
> | **Average** | **0.45** | **0.38** | **0.37** |
>
> We present the average head-CKA result for Llama, DIFFT and DEX (also trained from scratch), where lower score indicates "less redundancy" among the attention heads. It is clearly visible that both DIFFT and DEX show notably reduced attention head redundancy when compared to Llama model. We believe this enables both DIFFT and DEX to form richer representation that encodes a wider spectrum of information. Note that we compute pairwise head CKA with rbf kernel and average the scores for each layer.
>
> These three analyses consistently highlight the mechanistic similarity between DEX and DIFF, revealing shared properties in three key areas:
>
> 1. Selective filtering of attention noise;
> 2. Improved expressivity through negative attention; and
> 3. Reduced attention head redundancy.
>
> **Concern 2: The legends in the figures are small and hard to read.**
>
> We apologize for this. We humbly accept your feedback and we will make sure to enlarge the legends for our figures upon revision.
>
> **Question 1: Could the authors train for a longer duration and report these extended results to explore whether a regular Transformer architecture might ultimately be superior?**
>
> The original result that compares Differential Transformer and Llama model is introduced in **Tab.8**. To answer your question, we have run continued pretraining on the four models (Llama, DIFF, DEX, DEX-from-scratch) for additional 10k iterations (approximately 6B train tokens). We present the average scores for 11 natural language tasks used in **Tab.1** that includes Arc-C, Arc-E, BoolQ, Hellaswag, etc.
>
> | Model | Average (Tab.8) | Average (+10k steps) |
> | :--- | :--- | :--- |
> | Llama | 42.4 | 42.5 |
> | DIFF | 42.9 | 42.9 |
> | DEX | 42.8 | 42.8 |
> | DEX-scratch | 42.8 | 42.9 |
>
> We can see that additional training gives a slight improvement on the downstream scores for Llama and DEX-scratch, but the change is insufficient to close the performance gap with DIFF. Nevertheless, we believe it was a highly relevant question to check the final performance after additional training, given the trend in Fig.8a.
>
> We once again thank you for your insightful and constructive review. We have provided additional explanations and experiments which we believe address your concerns. We hope our response resolves these points and further strengthens your positive assessment of our work. Thank you.

---

> > ### Comment · Reviewer_HPZ6 · 2025-08-04
> >
> > Thank you for your response. I think the authors addressed most of my concerns, and I will maintain my (positive) score.

---

> > > ### Author Response · Authors · 2025-08-05
> > >
> > > We are glad to be able to address your concerns. We deeply appreciate your insightful review, and let us know if you have any further questions or comments. Thank you.

---

### Official Review · Reviewer_WJux · 2025-06-30

**Clarity:** 3
**Significance:** 3
**Originality:** 2
**Rating:** 4
**Confidence:** 3

**Summary:**

The paper analyzes why Differential Transformers (DIFF) outperform standard transformers and proposes DEX, a lightweight, efficient extension for infusing differential attention's benefits into existing pretrained models without the need for training from scratch. The core claims are that DIFF attention enables negative attention scores (higher expressivity), reduces attention head redundancy, and improves learning dynamics. DEX is evaluated on various benchmarks, demonstrating improved performance with minimal data and computational overhead

**Questions:**

Please see weaknesses.

**Ethical Concerns:**

["NO or VERY MINOR ethics concerns only"]

**Final Justification:**

The author has addressed my concerns about the custom dataset used, the lack of comparison with DIFFT, the robustness analysis on head selection. I do think their reply is detailed enough. While I appreciate their efforts to ensure a fair comparison given the limited computing resources, I still believe that additional large-scale experiments for DIFFT and DEX would further strengthen their conclusions regarding the significance of DEX, and should be included in the revised version. Given that, I increase my score to 4.

**Limitations:**

Yes

**Paper Formatting Concerns:**

No formatting issues found.

**Quality:**

3

**Strengths And Weaknesses:**

Strengths:
- Provide detailed insights to the effective of Differential Transformers.
- Propose DEX - a novel lightweight alternative for DIFFT, which can leverage minimal data and computational overhead.
- Experiments are conducted on diverse benchmarks, showcasing the benefits of DEX.

Weaknesses:
- My main concerns lie on the experimental setup. Current experiments use a **custom-mixed dataset**, not the original pretraining data for models like Llama or Qwen. Thus, the absolute improvement and generalization to other training regimes or more realistic finetuning settings are somewhat unclear. Besides, **direct comparisons to original DIFF Transformer are limited**. The authors should provide the performance of DIFF on the main language modeling benchmark (different models and scope), and other tasks (Key IR, In-Context Learning).

- **Ambiguity in Head Selection Strategy**: The paper proposes two head selection strategies—low importance and high entropy—but lacks detailed comparative results clearly indicating why one strategy is preferable. Additionally, practical guidelines or robustness analyses concerning the selection of heads across different tasks and models would improve usability.

- **Limited Theoretical Justification**: The paper utilizes empirical observations and visualizations (e.g., attention entropy, redundancy) to interpret the Differential Transformers. However, it provides limited theoretical insights to justify why the differential attention approach outperforms standard attention mechanisms beyond empirical observations. A deeper theoretical exploration or formal analysis would significantly enhance understanding.

- **Potentially Overstated Generalization Claims:** The claim of significant gains using minimal adaptation data (<0.01%) is strong but could be overstated. Further analysis is needed regarding whether such minimal training consistently delivers strong performance across a broader range of datasets and scenarios.

---

> ### Author Rebuttal · Authors · 2025-07-29
>
> We deeply appreciate your thorough review and constructive comments. We are glad to hear your positive assessment of our paper on "providing detailed insights", "proposing a novel method" and "conducting diverse experiments". We are happy to address your concerns through this rebuttal.
>
> **Concern 1a: Experiments use custom-mixed dataset, not the original pretraining data, which makes the absolute improvement unclear.**
>
> This is a very relevant feedback, and we agree that this setting can potentially complicate the evaluation. However, we would be grateful if you understand that the original pretraining dataset for Llama or Qwen models are **not publicly available**, necessitating the use of custom dataset. Therefore, in order to minimize the confounding effect of the train dataset, we ran all other baselines (full tuning, LoRA, etc) on the exact same corpus. Full fine-tuning (FT) corresponds to 'continued pretraining' that maximally fits the model to the additional training data, while LoRA represents more robust finetuning method at the other end of the spectrum that preserves knowledge from the original training data. The fact that DEX consistently outperforms both methods testifies for its fundamental advantage. We ran comprehensive experiments using 5 model variants and 11 benchmarks, and DEX has delivered robust performance gains across most settings.
>
> Also, **Tab.7** presents DEX applied to instruction tuning setting, where we start from a Llama-3.1-8B model checkpoint instruct-tuned on OpenHermes-2.5 dataset (open-source) then apply DEX and other baselines using the exact same OpenHermes-2.5 dataset. This further eliminates the effect of training data gap, robustly demonstrating the advantage of DEX.
>
> **Concern 1b: Direct comparison to original DIFFT (language modeling, Key IR, ICL) is missing.**
>
> We believe this is a valid feedback. First, we want to gently mention that we presented 11 downstream natural language modeling benchmark scores in **Tab.8**, where we compare the performance of Llama model, DIFFT, DEX from scratch, and DEX from Llama checkpoint (our proposed adaptation setting) all trained on the exact same dataset (see Appendix F) along with qualitative CKA results (**Fig.19**). To further address your concerns, we ran additional evaluations on key information retrieval (KIR) and in-context learning (ICL) for these models, and present the results below:
>
> | KIR | 256 | 512 | 768 | 1024 | 1280 | 1536 | 1792 | 2048 | Avg |
> |---|---|---|---|---|---|---|---|---|---|
> | Llama | 0.09 | 0.05 | 0.04 | 0.06 | 0.11 | 0.04 | 0.02 | 0.06 | 0.059 |
> | DIFFT | 0.12 | 0.07 | 0.10 | 0.09 | 0.13 | 0.08 | 0.05 | 0.11 | 0.094 |
> | DEX | 0.11 | 0.13 | 0.08 | 0.12 | 0.08 | 0.07 | 0.07 | 0.08 | 0.093 |
> | DEX-S | 0.14 | 0.09 | 0.10 | 0.18 | 0.19 | 0.11 | 0.10 | 0.12 | **0.129** |
>
> | ICL | n=1 | 5 | 10 | 15 | 20 | 30 | 40 | 50 | AVG |
> |---|---|---|---|---|---|---|---|---|---|
> | Llama | 0.074 | 0.096 | 0.178 | 0.200 | 0.170 | 0.193 | 0.185 | 0.156 | 0.156 |
> | DIFFT | 0.074 | 0.141 | 0.148 | 0.156 | 0.207 | 0.193 | 0.207 | 0.207 | 0.167 |
> | DEX | 0.074 | 0.104 | 0.185 | 0.200 | 0.200 | 0.200 | 0.193 | 0.193 | **0.169** |
> | DEX-S | 0.067 | 0.119 | 0.185 | 0.193 | 0.185 | 0.193 | 0.200 | 0.193 | 0.167 |
>
> - DEX-S stands for DEX trained from scratch with the exact same training data as others.
> - Columns in KIR correspond to the context length, averaged across 5 different needle depths (0, 25, 50, 75, 100%) as in Fig.10.
> - Columns in ICL represent number of shots. We average the scores on three benchmarks used in Tab.3, Trec, Banking-77 and Clinic-150.
>
> It is evident from these results that DEX's performance is comparable to or often better than the original DIFFT (despite being much more efficient both during training and inference), consistently outperforming Llama.
>
> **Concern 2a: Detailed comparative results for head selection strategies are required.**
>
> Although we do have comparison for head selection strategies at **Tab 4**, we agree that supplementing experiments on this following your feedback would be beneficial. Hence, we present additional results below, under "concern 2b".
>
> **Concern 2b: Practical guidelines or robustness analyses on head selection across tasks and models would improve usability.**
>
> Including practical guidelines or robustness analyses is a great idea to improve the usability of our method in real life. We appreciate your constructive feedback, and share two relevant results from our additional experiments: (1) head selection across model variants and (2) head selection across downstream tasks.
>
> | Model | entropy | importance | all heads (no selection) |
> | :--- | :--- | :--- | :--- |
> | Llama-3B | **64.2** | 63.9 | 61.9 |
> | Llama-1B | **55.6** | 55.1 | 54.0 |
> | Qwen-1.5B | **62.4** | 62.0 | 61.6 |
> | Qwen-0.5B | **54.4** | **54.4** | 52.8 |
>
> - The scores are averaged across 11 widely used natural language benchmarks presented in Tab.1.
>
> | Task | entropy | importance | all heads (no selection) |
> | :--- | :--- | :--- | :--- |
> | Broad Knowledge | **63.1** | **63.1** | 62.5 |
> | Instruction Following | **57.2** | 55.4 | 54.9 |
> | Math | **74.3** | **74.3** | 70.6 |
> | Science | 77.7 | **77.8** | 75.8 |
>
> - We run this experiment with Llama-3.1-8B model instruct-tuned on Open-Hermes-2.5 dataset, as detailed in Appendix C.
> - We categorize MMLU, IFEval, GSM8k and Arc-C as broad knowledge, instruction following, math and science task, respectively.
>
> It is evident from these results that both head selection strategies perform robustly across different models and tasks (data), yielding notable improvements over the 'no selection' baseline. We would recommend using entropy-based selection for most practical uses, which delivers the best performance in most of the cases. We will include these results upon revision.
>
> **Concern 3: Theoretical justification for differential attention’s empirical advantage is limited.**
>
> Differential Transformer is a new architecture that empirically delivers strong performances. Conceptually, the introduction of DIFF will look like
>
> ”Different architecture” → “Different internal patterns/mechanisms” → “Different (better) downstream results.”
>
> However, the fact that little is known about the *“internal patterns/mechanisms”* of DIFFT makes this a “missing link” in our understanding of the framework, which motivates our work.
>
> Our main contributions are two-folds:
>
> 1. Provide insightful analyses that identify what internal mechanisms actually make differential transformers effective (expressivity of real-valued attention, head diversity, better learning dynamics)
> 2. Using the identified empirical characteristics as our inspiration, propose a data- and compute-efficient adaptation framework that injects DIFFT’s benefits (stronger language understanding, key information retrieval, in-context learning, etc) to pretrained LLMs without training from scratch (unlike the original paper)
>
> Unfortunately, providing theoretical justification for “why DIFFT *should* perform better” is out of our scope, though that will be an impactful topic for future research. We humbly ask for your understanding on this matter, and hope you find our contributions interesting and meaningful.
>
> **Concern 4: Analysis with a broader range of datasets and scenarios is needed to claim significant gains using minimal adaptation data.**
>
> In **Tab.1**, we show DEX outperforming baselines applied to 5 different models with <0.01% original training data. In **Tab.2-3** and **Fig.10**, we demonstrate that DEX can improve both key-information retrieval and in-context learning capabilities with minimal (<0.01%) training data. We believe our evaluation at this point is comprehensive, but might not be "general enough" as these results all belong to DEX applied to different “pretrained” models using additional “pretraining data”.
>
> In  **Tab.7**, we present two different scenarios: DEX applied to “pretrained” model using “instruction data” and DEX applied to “instruct-tuned” model using “instruction data” (continuous instruction tuning). We evaluate these scenarios using a wide variety of benchmark datasets spanning instruction following, scientific reasoning, math and coding, where DEX delivers impressive overall performances. We want to note that the training dataset for these experiments (Open-Hermes-2.5) spans roughly 350M tokens, which is much less than 0.01% of the 15T+ original training tokens for Llama-3 model [1]. Using DEX, it takes less than a day to tune an 8B model with a single A100 node, which further supports DEX's effectiveness under minimal training.
>
> Hence, we have provided empirical results on 5 models, 3 scenarios (pretrain+pretrain, pretrain+instruct, instruct+instruct), and over 20 benchmarks (11 language modeling, 6 instruction, 3 ICL, KIR, etc). We acknowledge that extensive evaluation is indeed needed to assess generalization, but we would be grateful if you kindly consider the depth and breadth of our evaluation for this matter.
>
> We once again thank you for your thorough and helpful review. We have provided additional explanations and experiments which we believe address your concerns. We hope our response resolves your concerns and steers your assessment of our work in a positive direction. Thank you.
>
> [1] Dubey, Abhimanyu, et al. "The llama 3 herd of models." arXiv e-prints (2024): arXiv-2407.

---

> > ### Comment · Reviewer_WJux · 2025-08-05
> >
> > I thank the authors for their detailed response, which has addressed most of my concerns. While I appreciate their efforts to ensure a fair comparison given the limited computing resources, I still believe that additional large-scale experiments for DIFFT and DEX would further strengthen their conclusions regarding the significance of DEX, and should be included in the revised version. That said, I will raise my score to Borderline Accept.

---

> > > ### Author Response · Authors · 2025-08-05
> > >
> > > Thank you for your response, and we are glad to have addressed most of your concerns. We agree that demonstrating the effectiveness of DEX against DIFFT at a larger scale would significantly strengthen the paper. We commit to including these larger-scale experiments in the revised version to validate DEX's scalability. Thank you again for your thoughtful and constructive comments.

---

### Official Review · Reviewer_WqWH · 2025-07-02

**Clarity:** 3
**Significance:** 3
**Originality:** 4
**Rating:** 5
**Confidence:** 4

**Summary:**

Differential transformers have recently established itself as an architecture with competitive or better performance compared to traditional transformers. This paper provides an in-depth analysis of the benefits and working mechanism of differential transformers. By showing the main benefits of differential transformers, namely 1) negative attention scores potentially provides additional expressivity, 2) it allows focusing mainly on salient tokens, 3) the additional lambda parameter helps smoother learning dynamics.

The paper then goes on to provide a method called DEX that introduces the benefits of differential transformers directly on top of pretrained transformers. Dex uses a learnable scalar lambda and additional learnable projection matrix for a fraction of all attention weights in the pretrained transformer. This allows finetuning of pretrained transformers to get benefits of differential transformers.

Experiments on Llama-3.1-8B, Llama-3.2-3B/1B and Qwen-2.5-1.5B/0.5B finetuned on dataset created by the authors show clear benefit of DEX. Further experiments on key information retrieval and in-context learning shows the benefit of DEX.

**Questions:**

How important is the head selection, what happens if we do not use head selection in Tab. 4 and try out different combination of lambda_learned and lambda_annealing?

Please also check the weaknesses if it is possible to show experiments on other datasets

**Ethical Concerns:**

["NO or VERY MINOR ethics concerns only"]

**Final Justification:**

The authors have addressed the main limitation I had raised regarding older datasets: the authors have pointed out additional experiments that were present in the appendix as well as added additional experiments.

The authors have also added ablations regarding my question head-selection.

**Limitations:**

yes

**Paper Formatting Concerns:**

No concern

**Quality:**

3

**Strengths And Weaknesses:**

Strengths
- The paper provides an in-depth analysis of the advantages of differential transformer compared to a standard transformer, that is 1) more expressive attention weights since attention can be negative, 2) selective attention heads for salient tokens, 3) an additional lambda parameter that helps with the learning dynamics.

- The paper provides a method to obtain the benefits of a differential transformer on top of standard transformer by finetuning pretrained models. This allows preserving the pretrained knowledge as well as gain the benefits of differential transformers

- The paper also creates a dataset for finetuning that shows good results on the experiments

- Experimental results show improvement compared to standard finetuning of transformers

Weaknesses
- The experiments are restricted to older datasets. The models considered finetuned have been extensively evaluated on many recent datasets such as MATH500, GSM8k, MBPP, HumaEval, AIME (maybe AIME is not recommended for the selected models in the paper). Can the authors show performance on these datasets as well? The authors can also choose to finetune or apply their method on NuminaMath and evaluate on MATH500 or GSM8k (for non-overlapping cases between NuminaMath and Math500).

---

> ### Author Rebuttal · Authors · 2025-07-29
>
> Thank you for your helpful feedbacks and comments. We greatly appreciate your positive assessment about our work providing "in-depth analysis of differential transformer", proposing an effective method to "obtain the benefits on top of standard transformers" and showing strong experimental results. We hereby address your concerns in details.
>
> **Concern 1: Evaluation result on more recent benchmarks like MATH500, GSM8k, MBPP and HumanEval is needed.**
>
> We believe this feedback is very relevant and insightful. We first want to gently mention that empirical results for the latest benchmarks such as GSM8k, MBPP++, AGIEval, IFEval, MMLU are presented in **Tab.7**, for both cases where DEX is applied to base pretrained checkpoint (Llama-3.1-8B) and instruction-tuned checkpoint (Llama-3.1-8B tuned on Open-Hermes-2.5 data). It is clearly visible that DEX significantly outperforms LoRA or full fine-tuning baselines across different scenarios.
>
> To further alleviate your concerns, we ran additional evaluations for MATH-500 and HumanEval with our checkpoints used in Tab.7. We present the extended results below.
>
> | Model | MMLU | ARC-C | IFEVAL | MBPP++ | GSM8K | AGIEval | HumanEval | Math500 | AVERAGE |
> | :--- | :--- | :--- | :--- | :--- | :--- | :--- | :--- | :--- | :--- |
> | **(1) Instruction-tuned** | | | | | | | | | |
> | Base | 62.9 | 78.3 | 46.8 | 68.3 | 71.1 | 32.2 | 44.5 | 13.4 | 52.2 |
> | + LoRA | 63.1 | 79.5 | 45.7 | 65.3 | 70.3 | 40.6 | 47.0 | 4.0 | 51.9 |
> | + FT | 63.0 | 78.6 | 49.2 | 63.2 | 76.5 | 42.3 | 36.6 | 20.2 | 53.7 |
> | + DEX | 63.1 | 77.7 | 57.2 | 64.8 | 74.3 | 40.7 | 47.6 | 19.2 | **55.6** |
> | **(2) Pretrained** | | | | | | | | | |
> | + LoRA | 63.7 | 70.5 | 42.0 | 65.3 | 57.4 | 35.4 | 45.7 | 2.0 | 47.8 |
> | + DEX | 63.6 | 77.3 | 51.0 | 66.1 | 68.4 | 37.9 | 50.6 | 16.2 | **53.9** |
>
> - The training with Open-Hermes-2.5 instruct data is also very lightweight, taking less than a day to tune an 8B model with a single A100 node. The number of training tokens roughly sums to 350M, which is much less than 0.01% of the 15T+ tokens of llama-3 model [1].
> - Please refer to Appendix C for full details.
>
> It is clear that DEX delivers robust performance across a wide variety of scenarios and tasks, including broad knowledge (MMLU), instruction following (IFEval), scientific reasoning (Arc-C), math (GSM8k) and coding (MBPP++, HumanEval). It is noteworthy how DEX outperforms both heavier tuning method (FT) and lighter tuning method (LoRA), showing its fundamental advantage as an adaptation method.
>
> **Question 1: What happens if we try out different combinations of lambda learned and lambda annealing without head selection in Tab.4?**
>
> This is an interesting comment as there are many different possible combinations for our framework. We present results from the additional experiments to address your question.
>
> | Combination | all heads (no selection) | entropy-based selection |
> | :--- | :--- | :--- |
> | +learn, +anneal | 61.9 | 64.2 |
> | +learn, -anneal | 61.7 | 63.8 |
> | -learn, +anneal | 61.6 | 63.4 |
> | -learn, -anneal | 61.3 | 62.4 |
>
> - We run these experiments using Llama-3.2-3B model.
> - We report the average score for 11 natural language tasks shown in Tab. 1.
> - For all these configurations, we initialize the lambda value using the original DIFFT scheme, and control each component.
>
> It is clear that combining both "learnable lambda" and "lambda annealing" yields the best performance regardless of the head selection.
>
> We once again thank you for your insightful review. We have provided additional explanations and experiments which we believe address your concerns. We hope our response resolves your concerns and further strengthens your positive assessment of our work. Thank you.
>
> [1] Dubey, Abhimanyu, et al. "The llama 3 herd of models." arXiv e-prints (2024): arXiv-2407.

---

> > ### Author Response · Authors · 2025-08-06
> >
> > Dear Reviewer WqWH,
> >
> > With the discussion period ending in three days, we are writing to see if we can offer any further clarification on our rebuttal. We hope that our response and the additional results have successfully addressed your concerns. We would be grateful for the opportunity to provide any further clarifications. Please let us know if you have any remaining questions or concerns.
> >
> > Thank you again for your time and valuable feedback.

---

### Official Review · Reviewer_9JJa · 2025-07-02

**Clarity:** 2
**Significance:** 3
**Originality:** 3
**Rating:** 4
**Confidence:** 4

**Summary:**

The paper introduces DEX (Differential Extension), an extension designed for the recently proposed Differential Transformer (DIFFT) framework.  The paper begins by explaining why DIFFT works in practice, empirically demonstrating that DIFFT takes advantage of: a) discrepancies between all attention (vs salient) scores, (b) negative relevance in QK values, (c) reduced redundancy in attention heads, and (d) improved learning dynamics.  DEX is then introduced, the chief innovation of which is moving trainable parameters from a second set of QK matrices (i.e., DIFFT) to the regular output matrix (OV) minus a \lambda* parameterized-weight matrix *OV.  DEX also incorporates adaptive head selection (low-importance and high-entropy) during differential adaptation and an annealing schedule to \lambda.  DEX performance is subsequently demonstrated over Llama3 and Qwen2.5 models of various sizes against various PEFT methods (adapted over the base model) for common reasoning benchmarks.  While improvements are not always substantial, DEX does consistently improve performance across all models, which is not the case for competitor methods (including full fine-tuning).  Additional experiments key information retrieval and ICL also demonstrate the utility of the proposed method relative to full fine-tuning/LoRA/base model performance.

**Questions:**

# Minor suggestions
For Figure 1, could the authors add up arrows to subfigures (a) and (b) and downarrows for subfigures (c) and (d) (as well as mention these trends for the respective metrics))?  This will help the reader quickly parse the results (especially since the similarity trends in (a-b) are flipped in (c-d)) to see the stated conclusion: " It clearly shows that the overlap between the two attention scores is much greater in non-salient tokens."  Also, listing curve averages and contrasting "Top 5%" vs "All" will help quantify the stated conclusion.

# Questions
- "Although DEX is not presented as a fine-tuning method, we compare against baselines trained on the same data" <- Is DEX trained using full precision then?
- Is there a downside to Equation 3?  It seems like there is a tradeoff between granularity for efficiency that could be mentioned.

**Ethical Concerns:**

["NO or VERY MINOR ethics concerns only"]

**Final Justification:**

## Update during rebuttal
The authors have pointed to various points in both the paper and appendix to address much of my questions, with additional experiments demonstrating the better throughput achievable with DEX.  Furthermore, the authors have agreed to writing edits which quickly address these clarity issues in the main paper.

**Limitations:**

Yes

**Quality:**

3

**Strengths And Weaknesses:**

# Strengths
## Signifiance and Originality
The DIFFT framework has been a high impact contribution since its debut.  The fundamental idea behind DEX, i.e., removing dual QK matrices in favor of a single parameterized weight on the output-value matrix, is a very interesting and intuitive approach.

## Quality
Many of the experiments in Sections 2 and 4 were high quality, supporting the several points made in the text.

# Weaknesses
## Quality
A critical issue is the lack of necessary experiments/benchmarks.  E.g.:
> Direct comparison with original DIFF Transformer is limited by unavailable pretrained weights, which
 we defer to Appendix along with other details.

Comparison to DIFFT is necessary to show demonstrate the effectiveness of DEX relative to the work it builds on.  Even if the checkpoint is unavailable,
 the code from the Differential Transformer paper is available:
https://github.com/microsoft/unilm/tree/master/Diff-Transformer

Furthermore, given the custom data used for the experiments in Table 1, comparing to the DIFFT checkpoint would not be a fair comparison; to ensure a fair comparison, DIFFT would have to be trained on the same data and training recipe as used for DEX.

Additionally, one of the major claimed drawbacks to DIFFT is the following:
> We note that despite DIFF attention having the same number of parameters, it exhibits significantly higher compute cost and peak memory usage in practice due to enlarged dimensions

This is a main criticism of DIFFT, and motivation for DEX.  Thus, a heads up comparison showing both training/testing runtime and memory usage between DIFFT and DEX is required.

Finally, some presented experiments do not accurately support the claims being made.  E.g.:

"Training statistics further corroborate this finding. Fig.8 plots the language modeling loss and gradient
norms for the standard and DIFF transformer. While DIFF consistently achieves lower loss and more
stable grad norms, removing the learnable λ notably impairs optimization. We hypothesize that the
learnable λ plays a key role in stabilizing training dynamics, especially during the early stages." <- At the end of the day, these do not equate to downstream performance.  Could the authors test the model (is this Llama3?  Which one?) w and w/o learnable \lambdas on natural language tasks to show that there is some performative difference?

In Section 3.3, zero-initializing \lambda is the not the only option.  It is unclear whether annealing is required, or some randomly (perhaps Gaussian) sampled values would be sufficient.  Furthermore, in the ablation experiments:
> However, adopting the initialization scheme from the original DIFF Transformer setting yields slightly the best
performance.

Thus, it is thus unclear whether \lambda annealing is thus a valid design choice.  Based on these results, \lambda annealing is suboptimal and would ideally be replaced by the DIFFT initialization scheme in the main results.

## Clarity
The paper has substantial room to improve in clarity.  For instance, important details are lacking when describing the experiments.  These critically limit the ability to understand these experiments.
- For Figure 1, can the authors please detail the experiments either in the figure caption or inline (i.e., lines 70-79).  Reading through those lines, it is not clear what dataset or models are being presented.
- For Figure 2, could the authors please include what LLMs are being evaluated, e.g., is it Llama-3.2-1B?
- "Notably, this benefit is largely lost when the learnable λ is removed" <- What does this mean?  Is \lambda set to one in this case?
- Need similar model/data details for the following experiments: "Qualitative examples in Fig.4, such as down-weighting irrelevant subject in Indirect Object Identification task [77] or non-literal interpretation in sarcasm detection, illustrate how DIFF attention can achieve a more refined information flow using negative attention (green boxes)."
- Can the algorithms for lines 173-175 and 176-179 be written out for full clarity?
- "Figure 5" <- Please include a colorbar.  Also suggest adding a down arrow to (a) and an up arrow to (b).

The writing also has significant room for improvement.  Currently, the paper feels crammed, which may be why many details are lacking for the experiments in Section 2.  Some of the text could be tightened up/condensed, e.g., lines 145-151 and 152-158, as well as lines 161-164 (which would ideally go in the discussion section).

---

> ### Author Rebuttal · Authors · 2025-07-29
>
> Thank you for your thoughtful review and constructive feedbacks. We are glad you found our work interesting, original and containing high quality experiments. We will address your concerns one by one.
>
> **Concern 1: Comparison to the original Differential Transformer is missing.**
>
> We agree that comparison between our method (DEX) and DIFFT is an important part of our work. We want to gently mention that we presented 11 natural language benchmark scores in **Tab.8**, where we compare the performance of Llama, DIFFT, DEX from scratch, and DEX from Llama checkpoint (our proposed adaptation setting) all trained on the *exact same data* (see Appendix F.2) along with qualitative CKA results (**Fig.19**). We placed these results in the Appendix due to page limits, but we acknowledge that this choice has made these critical results less accessible to readers. We appreciate your constructive feedback, and promise to rearrange the tables to improve their visibility upon revision. We would appreciate if you kindly understand that we had to run these *from-scratch* pretrainings in smaller scales due to compute resource constraints (model and data specs in Appendix F).
>
> To supplement our comparative study and further address your concerns, we present key information retrieval (KIR) and in-context learning (ICL) evaluations below:
>
> | KIR | 256 | 512 | 768 | 1024 | 1280 | 1536 | 1792 | 2048 | Avg |
> |---|---|---|---|---|---|---|---|---|---|
> | Llama | 0.09 | 0.05 | 0.04 | 0.06 | 0.11 | 0.04 | 0.02 | 0.06 | 0.059 |
> | DIFFT | 0.12 | 0.07 | 0.10 | 0.09 | 0.13 | 0.08 | 0.05 | 0.11 | 0.094 |
> | DEX | 0.11 | 0.13 | 0.08 | 0.12 | 0.08 | 0.07 | 0.07 | 0.08 | 0.093 |
> | DEX-S | 0.14 | 0.09 | 0.10 | 0.18 | 0.19 | 0.11 | 0.10 | 0.12 | **0.129** |
>
> | ICL | n=1 | 5 | 10 | 15 | 20 | 30 | 40 | 50 | AVG |
> |---|---|---|---|---|---|---|---|---|---|
> | Llama | 0.074 | 0.096 | 0.178 | 0.200 | 0.170 | 0.193 | 0.185 | 0.156 | 0.156 |
> | DIFFT | 0.074 | 0.141 | 0.148 | 0.156 | 0.207 | 0.193 | 0.207 | 0.207 | 0.167 |
> | DEX | 0.074 | 0.104 | 0.185 | 0.200 | 0.200 | 0.200 | 0.193 | 0.193 | **0.169** |
> | DEX-S | 0.067 | 0.119 | 0.185 | 0.193 | 0.185 | 0.193 | 0.200 | 0.193 | 0.167 |
>
> - DEX-S stands for DEX trained from scratch with the exact same training data as others.
> - Columns in KIR correspond to the context length, averaged across 5 different needle depths (0, 25, 50, 75, 100%) as in Fig.10.
> - Columns in ICL represent the number of shots. We average the scores on three benchmarks used in Tab.3, Trec, Banking-77 and Clinic-150.
> - Full detail in Appendix F.
>
> **Concern 2: Train/test efficiency and memory usage comparison is required.**
>
> This is a great feedback. While we do have test-time throughput comparison in **Fig.12** (with Llama-3B model) where DEX shows far better efficiency than DIFFT, we additionally show other efficiency benchmarks (on Nvidia A100 GPU).
>
> **Test run time.**
>
> | Throughput (tokens/sec) | 1024 | 2048 | 4096 | 8192 | 16384 | 32768 | 65536 |
> |---|---|---|---|---|---|---|---|
> | Llama | 23117.8 | 23236.7 | 22731.5 | 20777.9 | 18327.2 | 14804.0 | 10272.9 |
> | DEX | 21787.5 | 22131.6 | 21775.2 | 20048.2 | 17788.0 | 14470.3 | 10127.4 |
> | DIFFT | 19474.2 | 18986.2 | 16972.2 | 13693.5 | 9766.9 | 5879.1 | 2757.1 |
>
> | Latency (ms) | 1024 | 2048 | 4096 | 8192 | 16384 | 32768 | 65536 |
> |---|---|---|---|---|---|---|---|
> | Llama | 44.3 | 88.1 | 180.2 | 394.5 | 894.4 | 2213.5 | 6379.5 |
> | DEX | 47.0 | 92.5 | 188.1 | 408.8 | 921.5 | 2264.5 | 6471.1 |
> | DIFFT | 52.6 | 107.9 | 241.3 | 598.5 | 1678.3 | 5573.6 | 23770.0 |
>
> | peak Vram (GB) | 1024 | 2048 | 4096 | 8192 | 16384 | 32768 | 65536 |
> |---|---|---|---|---|---|---|---|
> | Llama | 6.49 | 6.98 | 7.97 | 9.96 | 13.92 | 21.84 | 37.68 |
> | DEX | 6.61 | 7.11 | 8.10 | 10.08 | 14.04 | 21.96 | 37.80 |
> | DIFFT | 6.49 | 6.98 | 7.97 | 9.96 | 13.92 | 21.84 | 37.68 |
>
> - Above tables measure test-time throughput, latency and peak memory usage with varying context lengths (column) and batch size set to 1.
> - Note that DEX uses additional 120mb peak memory (constant regardless of input) due to the learnable projection parameters.
>
> **Train run time.**
>
> | Train | sec/step | total runtime | peak VRAM |
> |---|---|---|---|
> | Llama | 1.42 | 3h 33m | 68.1 |
> | DEX | 1.52 | 3h 50m | 68.8 |
> | DIFFT | 2.52 | 6h 18m | 70.7 |
>
> - 0.4B model (Appendix F), 890B training tokens, 24k context window, AdamW optimizer, 4 gpus, bf16 automatic mixed precision, pytorch SDPA implementation with no gradient checkpointing.
>
> **Concern 3a: Ablation for learnable \lambda's on natural language tasks is needed.**
>
> We would like to gently guide you to **Tab.4** (bottom 3 rows), where we present results with and without the learnable lambda on 11 natural language tasks (done on Llama-3B model). For additional clarity, we present full 11 benchmark scores for Llama-3B, Llama-1B and Qwen-0.5B model.
>
> | Model / Variant | Arc-C | Arc-E | BoolQ | COPA | Hellaswag | MNLI | OBQA | PIQA | WIC | Winogrande | WSC | AVERAGE |
> | :--- | :--- | :--- | :--- | :--- | :--- | :--- | :--- | :--- | :--- | :--- | :--- | :--- |
> | **LLaMA-3.2 3B** | | | | | | | | | | | | |
> | +learnable | 45.5 | 73.3 | 74.8 | 84.0 | 74.1 | 49.5 | 42.6 | 78.2 | 51.9 | 69.1 | 63.5 | **64.2** |
> | -learnable | 45.5 | 73.0 | 71.3 | 83.0 | 73.4 | 49.6 | 40.8 | 77.6 | 53.6 | 67.4 | 62.5 | **63.4** |
> | **LLaMA-3.2 1B** | | | | | | | | | | | | |
> | +learnable | 35.2 | 64.2 | 57.8 | 79.0 | 64.0 | 38.0 | 38.0 | 75.0 | 51.9 | 60.6 | 48.1 | **55.6** |
> | -learnable | 34.5 | 63.1 | 54.5 | 79.0 | 63.8 | 43.3 | 37.4 | 75.0 | 53.0 | 60.3 | 36.5 | **54.6** |
> | **Qwen-0.5B** | | | | | | | | | | | | |
> | +learnable | 34.8 | 65.2 | 56.5 | 73.0 | 52.1 | 40.1 | 35.4 | 70.1 | 51.6 | 57.6 | 61.5 | **54.4** |
> | -learnable | 33.8 | 64.4 | 56.4 | 73.0 | 51.7 | 35.2 | 35.2 | 69.5 | 50.0 | 56.5 | 53.8 | **52.7** |
>
> - Here, we initialize lambda with DIFF scheme and apply lambda annealing, only changing learnable lambda to non-learnable.
>
> **Concern 3b: The effectiveness of \lambda annealing is unclear, especially with different \lambda initializations.**
>
> Although we present lambda-anneal ablation in **Tab.4** and lambda-initialization ablation in **Tab.5**, we believe that having experiments that put these two variables together would clarify the effectiveness of lambda-annealing. To address your concern, we report the following result:
>
> | Initialization | 0.8 | 0.5 | 0.3 | DIFF | N(0.5, 0.1) | Unif(0, 1) |
> | :--- | :--- | :--- | :--- | :--- | :--- | :--- |
> | +anneal | 54.3 | 54.0 | 54.2 | 54.4 | 54.3 | 54.1 |
> | -anneal | 54.1 | 54.0 | 54.1 | 54.2 | 54.0 | 53.9 |
>
> - 'DIFF' in column marks the init scheme from DIFF paper.
> - lambda is learnable.
> - We report average scores from 11 natural language benchmarks in Tab.1, where Qwen-0.5B model is used for efficiency. Same training data is used as explained in Sec.4.
>
> We can see that lambda annealing consistently delivers performance gains with different initializations.
>
> **Concern 4: The model and dataset being evaluated is unclear in Fig 1 and Fig 2.**
>
> We mention the model and dataset used for the analyses in Sec.2 in **L50-52**, with reference to full details in **Appendix F.2**. We use the validation split of our training data for the analyses.
>
> **Concern 5: Clarification is needed for the statement “Notably, this benefit is largely lost when the learnable λ is removed."**
>
> We apologize for the simplified explanation. Specifically, we initialize lambda using the original DIFFT scheme and fix it (non-learnable), thereby only removing the “learnable” part of lambda dynamics.
>
> **Concern 6: The model and dataset being evaluated is unclear in Fig 4.**
>
> The experimental setup for all analyses in Sec.2 is mentioned in **L50-52** and detailed in **Appendix F.2**. For IOI task, as the citation points to in **L96**, we use the data setting from the paper "Interpretability in the Wild: a Circuit for Indirect Object Identification in GPT-2 small". We will supplement explanation about the specific settings for further clarity.
>
> **Concern 7: Can the algorithms for lines 173-175 and 176-179 be written out for full clarity?**
>
> Absolutely, we will add the full algorithms in the Appendix for maximal clarity. Here, we provide brief sketches to alleviate your concerns.
>
> 1. Entropy-based
> - For each head ($h$) in layer ($l$), compute its average attention entropy scores $\bar{\mathcal{E}}_{l,h}$, which is the mean entropy taken over all samples (we pass in 50 samples from our train data).
> - In each layer $l$, we rank all heads based on their average entropy score.
> - We select the top-$k$ heads with the **highest** entropy scores from each layer's ranked list to be modified by DEX.
>
> 2. Importance-based
> - For each head ($h$) in layer ($l$), compute its head importance metric (using [57,14] in the main paper).
> - In each layer $l$, we rank all heads based on their importance metric.
> - We select the top-$k$ heads with the **lowest** importance scores from each layer as the target for DEX.
>
> **Concern 8: Adding colorbar, down arrow and up arrow can help improve Fig 5.**
>
> We will surely do that to improve the clarity of our figure.
>
> **Question 1: Is DEX trained using full precision?**
>
> We adopt mixed precision training using bf16, deepspeed-zero3 and AdamW optimizer.
>
> **Question 2: Is there tradeoff between granularity and efficiency for Eq.3?**
>
> Directly modifying the attention map (QK) would be a direct and "granular" way to manipulate the attention forward (as in DIFFT). However, this hurts efficiency either by damaging pretrained knowledge (more training is needed) or incurring more compute (slower inference), as mentioned in **L148-150**. We show that DEX, despite being efficient, brings the strengths of full DIFFT.
>
> We once again thank you for your thorough and constructive review. We have presented additional explanations and experiments which we believe address your concerns. We hope our response resolves your concerns and shifts your assessment of our work in a positive direction. Thank you.

---

> > ### Comment · Reviewer_9JJa · 2025-08-04
> > **Reply**
> >
> > Firstly, thank you to the authors for carefully clarifying all points in my review.  I agree, I can see the apples-to-apples comparison the authors have included with DIFFT.  Concerns with the scale of results in Appendix F.1 are largely mitigated by the extensive PEFT experiments in the paper.  The authors have agreed to add these clarifications in the main text, which would help allay this concern (even with the authors' response, it took some time bouncing around to understand and verify the various pieces of information).  Suggestions to quickly address this in the paper:
> >
> > > which we defer to Appendix along with other details.
> >
> > The appendix results don't necessarily need to be moved to the main paper, but please add a succinct description of the apples-to-apples experiments/results in the main paper.
> >
> > Additionally, I am assuming the Llama architecture used for the 0.4 B architecture is Llama-2/Llama-3, i.e., group-query attention, is that correct?  Could the authors additionally add this clarification in the paper, i.e., change:
> >
> > > (1) a standard transformer baseline using the Llama architecture (Llama)
> >
> > to
> >
> > > (1) a standard 0.4B transformer baseline using the Llama-3 architecture (Llama)
> >
> > I appreciate addressing in detail my other concerns, which have been addressed, and the proposed changes to the appendix.  Given the authors response and the additional paper clarifications, I will raise my score accordingly.

---

> > > ### Author Response · Authors · 2025-08-05
> > >
> > > Thank you for your response, and we are truly glad to be able to address your concerns. We will make sure to add concise explanations in the main paper to help readers understand the specific experimental settings, model architecture (yes, you are right about Llama-3 architecture with GQA) and the results with better clarity.
> > >
> > > We sincerely appreciate your constructive feedbacks, which we believe helped us improve our work greatly. Thank you.

---

### Note · Authors · 2025-08-11

Differential Transformer (DIFFT) is an impactful work that introduces (a) a novel architecture that learns (b) a new set of internal mechanisms to deliver (c) better performance. However, little is known about (b), leaving a missing link between (a) and (c). Our work addresses this gap by identifying three distinctive patterns in DIFFT and leveraging these insights to design an efficient method that transfers its benefits to pretrained LLMs with a standard architecture.

We appreciate that our work has been recognized for **(1) providing detailed insights into the inner workings of DIFFT** (HPZ6, WJux, WqWH), **(2) proposing a novel and effective approach for efficient adaptation** (HPZ6, WJux, WqWH, 9JJa), and **(3) presenting diverse, strong, and high-quality experimental results** (HPZ6, WJux, WqWH, 9JJa).

The concerns raised in the initial reviews can be grouped into 3 categories, each of which we have addressed in our rebuttal:

> **Need for direct comparison to DIFFT** (9JJa, WJux, HPZ6)

- We clarified that direct comparisons on 11 popular language benchmarks (Tab.8) were presented, with full implementation details in Appendix F.2.

- Additional results on Key Information Retrieval and In-Context Learning show that DEX (ours) performs favorably against both standard Llama and DIFFT.

- Comparative analyses further demonstrate that DEX echoes the beneficial internal patterns of DIFFT.

> **Need for evaluation on more diverse downstream tasks** (WqWH, WJux)

- Our paper evaluates DEX on 5 models (Llama-1B, 3B, 8B, Qwen-0.5B, 1.5B) across 3 adaptation scenarios (PT+PT, PT+IT, IT+IT) over more than 20 benchmarks (natural language understanding, instruction following, broad knowledge, science, math, coding, in-context learning, key information retrieval).

- During rebuttal, we added results from MATH500 and HumanEval to further address this point.

> **Request for further ablation studies** (9JJa, WqWH, WJux)

- We clarified the existing ablations in Tab. 4, and provided extensive new studies on the learnable lambda, the annealing technique, and attention head selection strategies.

In summary, the initial concerns have been thoroughly addressed, both by clarifying results already present in the manuscript and by providing extensive new experiments during the rebuttal. We are gratified that the reviewers found our responses convincing during the discussion period and believe no major concerns remain. Thank you.

*PT: pretrain, IT: instruct-tune

---

### Decision · Program_Chairs · 2025-09-17

**Decision:**

Accept (poster)

**Comment:**

This paper investigates the Differential Transformer, whose strong empirical performance is often linked to noise-canceled attention but remains theoretically underexplored. The authors identify three factors behind its success–enhanced expressivity via negative attention, reduced redundancy among heads, and improved learning dynamics. Building on these insights, they propose DEX, a lightweight method that incorporates differential attention into pretrained language models by reusing softmax scores and applying a simple differential operation on the value matrix. DEX is efficient at both training and inference and achieves gains across diverse benchmarks.

While there are some limitations–most notably the absence of large-scale experiments for DIFFT and DEX–the reviewers agree that the paper’s in-depth analysis of differential transformers is insightful, and the proposed DEX method is novel. Since all reviewers have unanimously recommended acceptance with strong support from Reviewer WqWH, I likewise recommend acceptance of this paper.